# communications
# engineering

# Co-assessment of costs and environmental impacts for off-grid direct air carbon capture and storage systems

Moritz Gutsch [1,2] & Jens Leker [1,2]

Large-scale deployment of direct air carbon capture and storage (DACS) is required to offset $CO_2$ emissions. To guide decision-making, a combined assessment of costs and environmental impacts for DACS systems is necessary. Here we present a cost model and life cycle assessment for several combinations of off-grid DACSs, powered by photovoltaic (PV) energy and heat pumps combined with battery storages to mitigate intermittency of the PV energy source. Utilization factors of DACSs are estimated for different locations, power of PV systems and battery capacities. We find that the cost optimal layout for a DACS in Nevada (USA) with a nominal $CO_2$ removal capacity of 100,000t$CO_2$ per year consists of 100 MW PV and 300MWh battery. Costs are $755 and $877 for gross and net removal of 1t$CO_2$. The cost difference is explained by a carbon removal efficiency (CRE) of 88%. Of 16 evaluated environmental impact categories mineral resource use is most problematic. We conceive a dashboard which allows to track how changes to technical parameters, such as energy consumption or adsorbent degradation, impact costs, CRE and combined environmental impacts. In an optimized scenario and including tax credits, costs for net-removal of 1t$CO_2$ will be $216 at a CRE of 93%.

[1] Helmholtz Institute Münster, IEK-12, Forschungszentrum Jülich GmbH, Corrensstraße 46, Münster 48149, Germany. [2] Institute of Business Administration at the Department of Chemistry and Pharmacy, University of Münster, Leonardo Campus 1, Münster 48148, Germany. ✉email: m.gutsch@fz-juelich.de

Carbon dioxide ($CO_2$) is a problematic greenhouse gas (GHG) associated with global warming[1,2]. Achieving net zero $CO_2$ emissions is essential to stabilize the climate and doing so at an accelerated pace imperative to stay within the limit of 2 °C global warming compared to pre-industrial levels[3,4]. Direct air capture (DAC) is a promising technology which takes $CO_2$ out of ambient air using sorbents and subsequent recovery of highly concentrated $CO_2$. Depending on the overall system design, $CO_2$ captured by DAC units can be used in a utilization pathway (DACU) for production of chemicals and synthetic fuels[5] or permanently stored in the ground (DACS)[6]. While utilization of captured $CO_2$ does not count as $CO_2$ removal technology, DACS fulfills both principles, that is, $CO_2$ is captured from the atmosphere (principle 1) and stored durably, for example as rocks in the ground (principle 2)[7]. To capture $CO_2$ a flow of ambient air is forced across a high surface area of sorbent material, either on a solid framework or in a liquid solution[6,8]. Sorbent material saturated with $CO_2$ is then regenerated at low (100 °C) temperatures (LT) or high (>800 °C) temperatures (HT) depending on the system layout[6,8]. Technological readiness levels for both designs are between 6 (pilot plant) and 8 (small commercial scale)[8,9].

Net-zero estimates from the International Energy Agency (IEA) require DACS capacity of 85 Mt$CO_2$ / year in 2030 and almost 1 Gt$CO_2$ / year in 2050[10]. This is substantially up from an installed capacity of 8 kt$CO_2$ / year in 2021[10]. Powered by renewable energy sources DACS deployment is expected to substantially reduce climate-related health effects compared with scenarios that do not employ DACS[11], but problems associated with intermittency of renewable energy systems must be considered[12,13]. Several start-ups have introduced DACS systems on pilot or small industrial level and started purchase agreement with customers[8,14,15]. Further, in August 2023, the U.S. Department of Energy announced plans to spend $ 1.2 billion on the construction of DACS systems with a total capacity of 2 Mt$CO_2$ per year[16].

The large discrepancy between DACS plants in operation and the projected requirements of future development has led to a situation in which both costs and environmental impact assessments rely on projections and assumptions. As Climeworks charges customers $ 1200 per t$CO_2$ removed through its 4 kt system in Island, one indication about prices exists[17]. Still, prices do not have to reflect underlying costs and independent cost calculations are required[18]. Since the product DACS companies sell is 1 ton $CO_2$ removed from the atmosphere, any GHG emissions associated with the capture and removal, expressed by global warming potential (GWP), must be subtracted to get the amount of carbon credits which realistically can be sold at the market[13,18,19]. The carbon removal efficiency (CRE) is calculated according to Eq. 1 (refs. [19,20]).

$$CRE = \frac{m_{CO_{2,captured}} - GWP_{Capture\ process} - GWP_{Transportation\ \&\ storage}}{m_{CO_{2,captured}}}$$

(1)

Some earlier cost calculations have not accounted for reduced efficiency through low CREs[12], or included only direct emissions of the energy system[21–23], which are zero in case of photovoltaic and wind power but omit the fact that considerable GHG emissions were associated with the production of renewable energy systems (RES). Part of this problem has been addressed by reports from the IEAGHG[13] and IEA[10], by either conducting a cradle-to-gate life cycle assessment (LCA) for the impact category of climate change[13] or using LCA results from other work[10] to determine the CRE associated with capturing and storing 1 ton

$CO_2$. Two pressing issues associated with DACSs have yet to be addressed:

First, previous research on DACS has either focused on costs, for example refs. [12,21–25] or environmental LCAs, for example, refs. [20,26,27] (see also Supplementary Tables 1 and 2). The two IEA reports[10,13] included results from LCAs for the impact category of climate change to calculate CREs more accurately. Still, they did not assess other environmental impact categories next to costs, as should be done[28]. Consequently, a combined assessment of costs and environmental impacts of DACS is missing, a concern which has been shared in a recent literature review[9].

Second, the use of low-carbon energy sources for powering DACSs is crucial to achieve high CREs, providing potential for low net-costs. Ideally, low-carbon sources without intermittency problems, such as nuclear power, hydropower or geothermal power are used[18]. However, safety and societal concern about nuclear power or geographical distribution of hydropower and geothermal power are limiting factors. On the other side, wind power and photovoltaic (PV) are renewable energy sources which can be deployed more broadly, and forecasts estimate a rapid expansion of wind and PV capacity[29]. While intermittency of wind and PV systems presents a challenge, "autonomous" DACS powered by renewables hold promise. LT solid sorbent systems might be suited to deal with some intermittency[13]. In addition, LT systems put less requirements on water supply since cooling requirements are lower than for high-temperature systems, increasing the number of suitable locations[10]. Potentially, LT systems could even provide benefits as a local water source since water is co-extracted from the atmosphere alongside $CO_2$[10]. Main benefit of off-grid DACS is that the number of feasible locations is substantially expanded because restrictions for carbon-intensity of the local grid are avoided. Environmental promise of autonomous LT solid-sorbent DACS powered by PV and a lithium-ion battery (LIB) has been reported by Terlouw et al.[26] (though they caution that modelling of the photovoltaic system and battery could be improved). In cost calculations, variable RES have a strong effect on the load factor (utilization) of the DAC plant[10,12,13]. Since low utilization has adverse effect on the distribution of fixed costs to removed $CO_2$, costs for intermitted RES generally increase compared to a grid-connected system operating at full capacity[10,12,13]. Past cost estimates have taken a high-level perspective on intermittency problems and acknowledge that more work is required to model the RES more accurately to evaluate economic potentials of autonomous DACSs[10,13]. Therefore, a refined modelling of autonomous LT, solid-sorbent DACSs powered by intermitted PV-electricity is of interest (see Fig. 1).

Addressing both issues has motivated the present work. We develop a technological model for autonomous DACSs with a $CO_2$ capture capacity of 100,000 t$CO_2$ per year, powered by an off-grid PV system coupled with a LIB storage. In addition, we build on methods of cost modelling and LCA to conduct an integrated cost and environmental assessment using a shared technological basis, system boundary and functional unit. We identify promising layouts of the energy system, which is influenced by location, technical parameters, costs, and environmental impacts, for optimal autonomous DACSs.

## Results and discussion

**Estimating full load hours.** The DAC utilization model allows to estimate the hourly output of the DACS. With intermitted, renewable energy input, the DACS will not run at full capacity all the time. If the DAC runs at full capacity, it requires 22.6 MW electricity, of which 15.3 MW are used by heat pumps generating

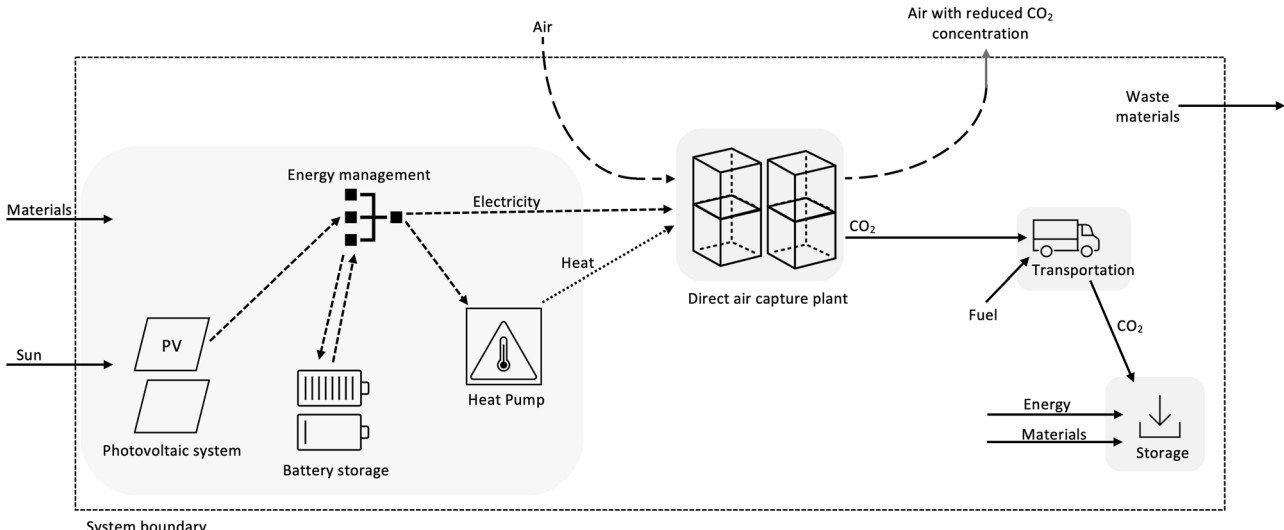

**Fig. 1 Layout of solid-sorbent, off-grid DACS powered by photovoltaic system, heat pump, and battery storage.** Electricity generated by the PV-system is used for powering the air fans and compression unit of the direct air capture plant. Thermal energy for sorbent regeneration is provided by heat pumps. The battery storage allows for a smoother operation of the DAC plant as intermittency problems of the photovoltaic system are (to some extend) addressed.

39.9 $MW_{heat}$ of thermal power. At full capacity the gross capture rate is 11.4 $tCO_2$ per hour, which is equivalent to 100,000 $tCO_2$ per year at 8760 operating hours. In Fig. 2, combinations of the energy system are denoted as "P-C" ("Power-Capacity") for power of the PV system in MW and capacity of the battery in MWh. Figure 2b indicates the operating profile of a 100 $ktCO_2$ per year off-grid DACS in Nevada (USA) using a 100 MW PV-system in combination with 100 MWh battery storage ("100–100"). Electricity generation is between zero and (almost) 100 MW. A distinct day-night relationship is present. If no battery was installed, operation of the DAC would be limited to hours of enough sun irradiation. With the battery, operation can be extended. Using a battery will also fill brief intermittency-gaps during the day to allow for a smoother operation of the DAC. Heat storage systems could also work but are excluded from the present analysis to limit complexity. Indicative examples are shown for the first 3 days of each quarter in 2020, but full year data is used to estimate utilization and output of the DAC plant. Average utilization in each of the four quarters stands between 6 and 7.5 $tCO_2$ per hour (50–65%) with a "100–100" system in Las Vegas (Nevada). For the whole of 2020, average utilization of the system is 57.3%, equivalent to 57,300 $tCO_2$ per year gross captured. Increasing the PV-power and battery capacity increases utilization of the DAC, since the turnoff time due to lack of energy is reduced. A relatively large capacity of the battery ensures operation until early morning hours, see Fig. 2a.

Las Vegas (Nevada) was selected due to the close location to suitable underground carbon dioxide storage, keeping transport distance to zero. Using solar irradiation data for Swakopmund (Namibia), which is similarly close to potential underground storage sides, leads to similar outcomes for the DAC utilization as Nevada. A slightly lower coefficient of performance (COP) for heat pumps used in Swakopmund, because of lower average temperatures than in Nevada, increases the energy consumption of the DAC system slightly. On the other side, a location in southern Germany (Munich) shows low average utilization of 46.4%, with sun irradiation in winter months unable to ensure a stable operation of the DAC.

As seen for the "200–300" combination in Nevada, utilization can approach 100% if enough excess PV power and storage capacity is installed. However, large energy systems are associated

with higher upfront investments, which could make the off-grid $CO_2$ removal prohibitively expensive. In addition, production of the energy system is associated with environmental impacts. Life cycle GHG emissions associated with the energy system lowers CRE which, in turn, increases costs for net $CO_2$ removal. Other environmental impacts do not directly translate to added costs but should be kept low in line with sustainable development principles.

**Annualized costs**. We start by looking at the annualized costs of the 100 $ktCO_2$ per year off-grid DACS system in Nevada. Due to a proportional relationship between the adsorbent degradation and amount of $CO_2$ captured[20,26,27], annual costs for adsorbent material also increase linear with the amount of $CO_2$ captured (both are variable costs). Annual depreciation (with annuity) does not depend on the utilization factor and has fixed cost character. Likewise, depreciation of the battery and photovoltaic system are fixed costs which depend on the layout of the energy system but not on utilization. Based on an iterative approach, numerous combinations of photovoltaic system and battery storage are evaluated. Power of the PV system ranges from 5 MW to 200 MW, capacity of the battery storage from 0 MWh to 300 MWh. Similar to Fig. 2, utilization of the DAC system is calculated for each combination (in Fig. 3 on the x-axis). In addition, we calculate the annualized costs of DACS (ACOD) of each system configuration, shown in Fig. 3 as grey dots. A given utilization is reached with several different energy system combinations. For example, both "55–80" and "150–50" systems lead to ca. 50% utilization or (50,000 $tCO_2$ / year gross removed). But, with annualized costs of $ 45.8 million, the "55–80" configuration is cheaper than the "150–50" combination with $ 54.4 million. Consequently, bars in Fig. 4 represent the lowest cost layout for utilizations between 5% and 96% (5000 to 96,000 $tCO_2$ per year removed).

The smallest energy system ("5–10") captures (and removes) ca. 5000 $tCO_2$ per year and is associated with ACOD of $ 26.6 million per year. Due to the low utilization, depreciation of the DAC plant accounts for $ 21.5 million or 81% of annualized costs. The highest cost-efficient, utilization (ca. 96,000 $tCO_2$ per year) is reached with a 185 MW PV and 300 MWh battery storage ("185–300"). Annualized costs are $ 80.8 million per year.

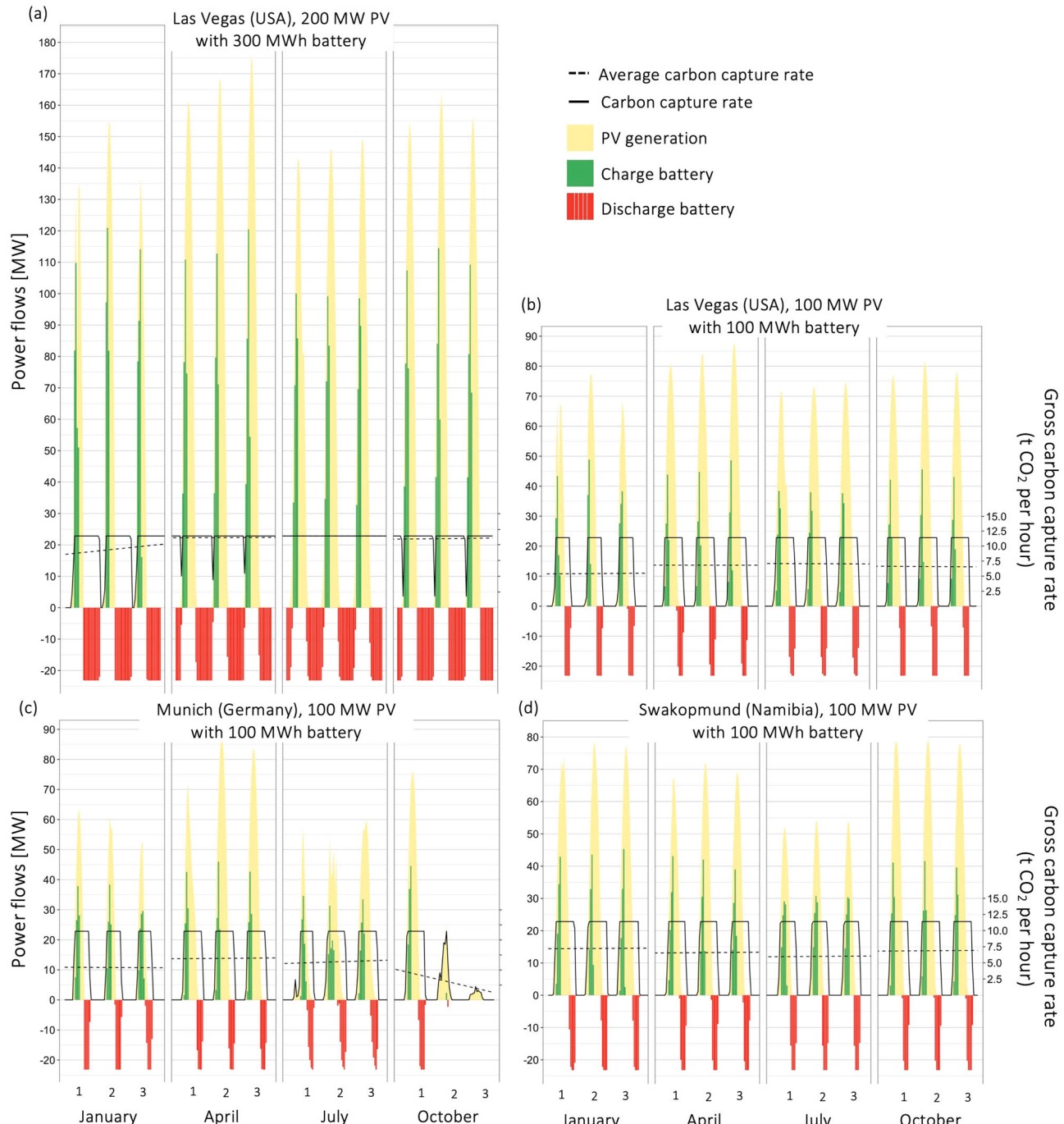

**Fig. 2 Estimating utilization factors of off-grid DACS with various combinations of energy system and location.** Electricity generated by the photovoltaic system (yellow) depends on the nominal power of the PV-system ("P") and the local solar irradiation profile. Excess electricity not required for powering the DACS system is used for charging the battery storage with nominal capacity ("C") (red), which is discharged if solar irradiation decreases (green) to provide energy for continued DACS operation. Whenever PV-system and battery are not sufficient to meet the power requirement of the DAC, the capture rate (tCO$_2$ per hour) is reduced. **a** DACS located in Las Vegas (USA) with PV-power of 200 MW and battery storage of 300 MWh. **b** Same location but smaller PV-system (100 MW) and battery (100 MWh). **c** DACS system with 100 MW PV-system and 100 MWh battery in Munich (Germany), and (**d**) Swakopmund (Namibia).

**Levelized cost.** Next, we divide annualized cost of DACS by the amount of annual gross CO$_2$ captured and removed to calculate the levelized costs of DACS (LCOD). Roughly speaking, the difference between the price that a DACS company charges customers must at minimum exceed levelized costs of its operation to avoid a loss. Our calculation shows lowest LCOD$_{gross}$ of \$ 775.2 for a "105–300" combination, see Fig. 4. Higher utilization, for example, 96% with a "185–300" combination increases LCOD$_{gross}$ to \$ 846.5. Compared to the "105–300" combination, depreciation of the DAC plant is reduced from \$ 236 to \$ 228 per ton CO$_2$ gross removed. However, depreciation of the substantially larger PV-system in a "185–300" system increases by \$ 85 per ton CO$_2$ compared to "105–300" combination. This increase more than offsets the reduced depreciation of the DAC plant and leads, in sum, to higher levelized costs.

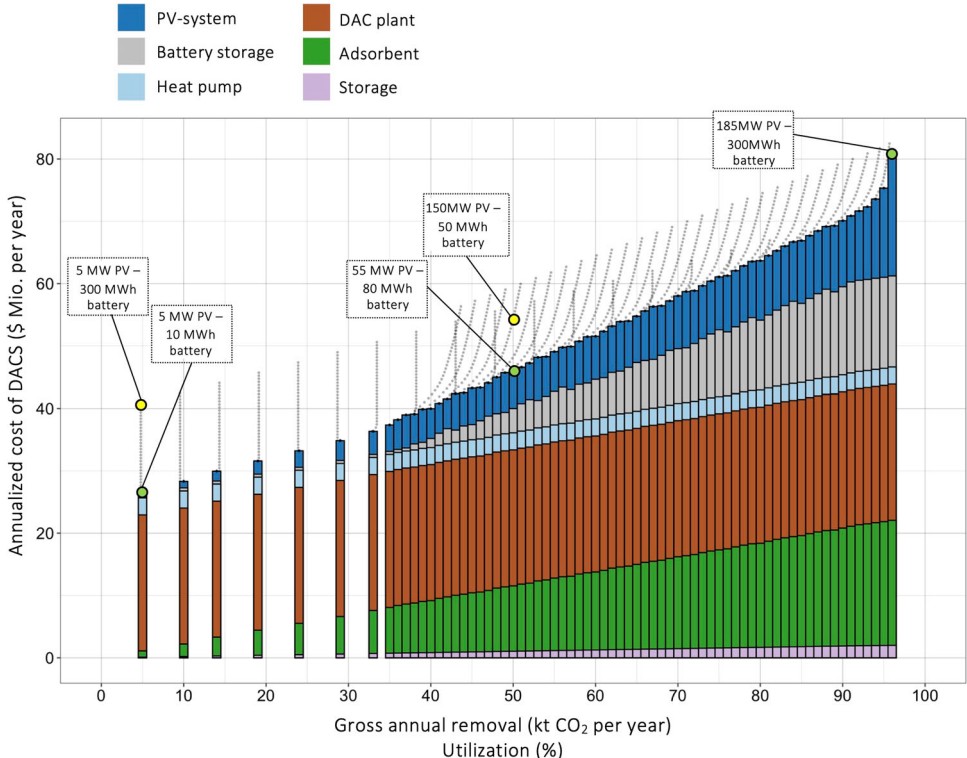

**Fig. 3 Annualized costs for off-grid DACS in Nevada (USA) with different combinations of photovoltaic system and battery storage.** Each dot represents the annualized costs ($ per year) for a different combination of power of the PV-system (from 5 MW to 200 MW) and battery capacity (0 MWh to 300 MWh). Bars represent the lowest-cost option for utilization factors between 0% and 100%, corresponding to an annual gross removal capacity between 0 ktCO₂ and 100 ktCO₂ per year. Black dots represent combinations of the energy system with comparable utilization factors as the bars but higher annualized costs. Colours within bars reflect the contribution of deprecation of PV-system, battery storage, heat pump, and DAC plant, as well as variable costs of adsorbent replacement and permanent underground storage of captured CO₂.

As present cost calculations are accompanied by life cycle assessments, we calculate GHG emissions associated with capture and removal of 1 ton $CO_2$ for each energy system combination (more in subsequent section). Dividing $LCOD_{gross}$ by the carbon removal efficiency, CRE, leads to $LCOD_{net}$, the levelized cost for net-removal of 1 $tCO_2$. The CRE is between 83.2% and 88.9% for annual outputs between 24,000 and 96,000 $tCO_2$ per year. For the lowest gross cost, "105–300" combination, a CRE of 88.3% brings $LCOD_{net}$ to $ 877.5. From a net-cost perspective, a "100–300" combination with a CRE of 88.5% achieves slightly lower $LCOD_{net}$ of $ 877.0, although $LCOD_{gross}$ are $ 0.8 higher than for the "105–300" combination. Recently, U.S. legislation has increased the tax credit for carbon capture with DAC systems to $ 180 per $tCO_2$ removed. Figure 4 includes this tax credit, with a small discount to match the longer lifetime of the DAC plant compared to the duration of the 45Q tax credit. Since credits are supposedly given for the net removal of $CO_2$, DACS systems with higher carbon removal efficiencies get slightly higher credits. Overall, credits lead to $ 724.2 as lowest costs for net removal of 1 $tCO_2$ ("100–300").

**Global warming potential**. As addressed in Fig. 4, the design of the energy system, that is, power of the PV-system and battery capacity, has implications for utilization and costs of $CO_2$ removal. The difference between costs for gross removal of 1 $tCO_2$ and net removal of 1 $tCO_2$, which accounts for life cycle GHG emissions associated with the capture and storage process, results from differing CREs. In Fig. 5, global warming potential associated with different DACS layouts is shown. For the net removal of 1 $tCO_2$ from the atmosphere, >1 $tCO_2$ must be

captured and removed through the DACS to offset emissions associated with the process and leakage of $CO_2$ from storage. Note that annual leakage is <0.01% in well-managed geological $CO_2$ storage sites[30].

GHG emissions associated with gross capture and removal of 1 $tCO_2$ are between 110.7 and 152.5 $kgCO_2eq$ (functional unit 1). Major contributions come from the PV system and the battery storage, underlining that even though the energy system does not release GHGs during operation, production of the system adds a GWP burden. Thus, for the net removal of 1 $tCO_2$ with a "100–300" system, a gross 1,115 $tCO_2$ must be captured and removed from the atmosphere of which the 0.115 $tCO_2$ offset the GWP of the process. In Fig. 5, GHG emissions are presented for lowest net-cost combinations. However, in some cases different energy systems are associated with lower GHG emissions than the low-cost combinations. This is expressed by red bars in Fig. 5, showing the difference between lowest cost and lowest GWP setups for similar utilization factors. For instance, a lowest net cost "45–20" combination achieves 113.5 $kgCO_2eq$ per $tCO_2$ at 38% utilization. A "40–30" combination, on the other side, achieves 110.9 $kgCO_2eq$ per $tCO_2$ at a comparable utilization of 38%. Generally, though, we find that differences between lowest net cost and lowest GWP combinations are small. The PV system and battery have a more substantial contribution to GWP than to costs. DAC plant and adsorbent material, on the other side, tend to have a higher contribution to costs than to GWP. In addition, we see a steep increase in GWP per ton $CO_2$ removed for high utilization (>95%), whereas low utilization of 20–30% are not associated with a steep increase. This contrasts with levelized costs in Fig. 4, where costs for low utilization (20%−30%) were much higher than costs for high utilizations >95%.

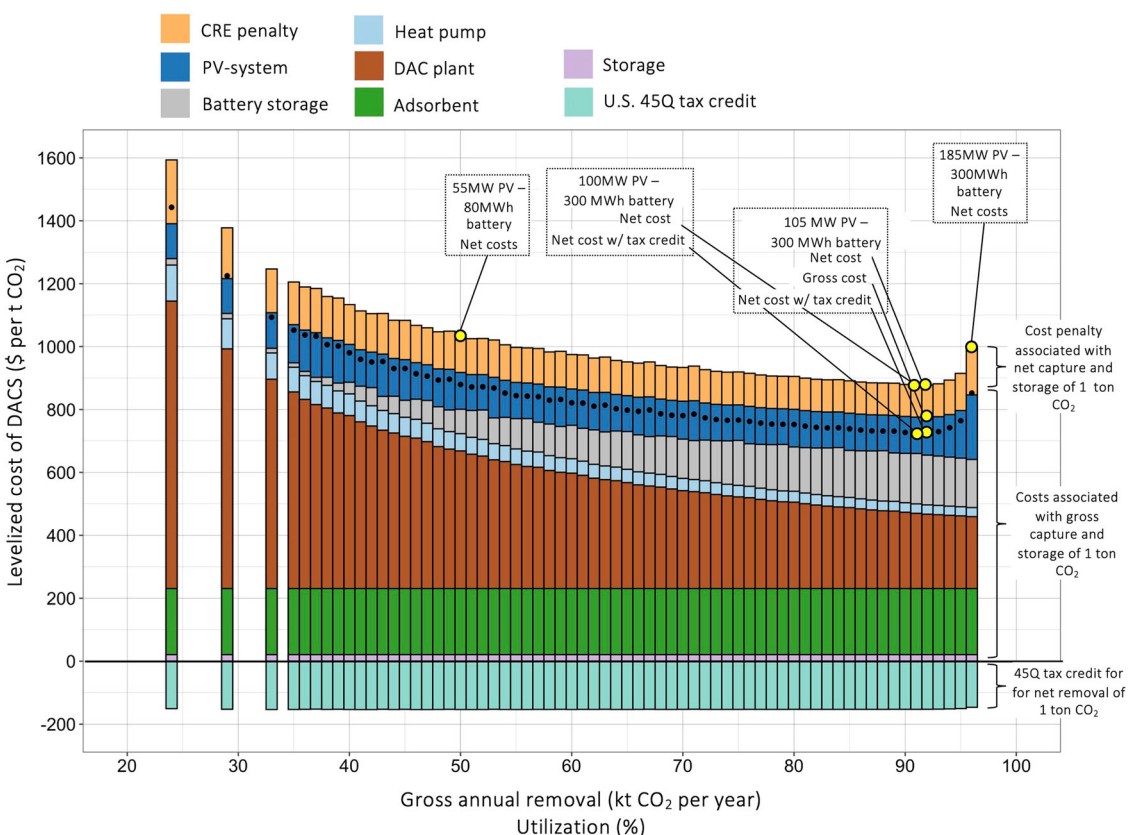

**Fig. 4 Levelized costs for gross and net removal of 1 tCO$_2$ with off-grid DACS in Nevada (USA).** Bars represent the lowest (net) cost configurations of the PV-system and battery storage for different utilization factors. Colours of the bars reflect the contribution of each component (depreciation of battery, etc.) to total costs for the permanent removal of 1 tCO$_2$. Yellow parts of bar highlight the difference between costs for gross and net removal of 1 tCO$_2$. Costs for net removal of 1 tCO$_2$ include the carbon removal efficiency (CRE), which is <100% for each configuration. Black dots represent the costs for net removal of 1 tCO$_2$ if the 45Q tax credit is subtracted from net costs.

**Normalized and weighted environmental impacts.** Apart from climate change, 15 other environmental impact categories are included in the life cycle assessment. To put the environmental impact categories into broader context, a normalization is done against the annual global impacts in each environmental impact category[31]. Further, relative weights are assigned to each impact category so that one final score is obtained for comparison of different options. Note, however, that the weights provided by the European Commission[31] are subjective and do not yet represent an agreed upon standard. Figure 6 shows normalized and weighed environmental impacts for the net removal of 1 tCO$_2$ with a 100 MW PV and 300 MWh battery storage in Nevada. Environmental benefits of $-3.8 \times 10^{-12}$ points are achieved in the impact category of climate change, which aligns with the general idea of DACS as a negative emissions technology. On the other side, each ton CO$_2$ removed from atmosphere brings environmental burdens in other impact categories. Mineral resource scarcity, mostly associated with mining of critical raw materials used for the battery storage and PV system, but also adsorbent material and DAC plant, is of concern. An environmental burden of $6.0 \times 10^{-12}$ points result for the "100–300" DACS layout. Put into context, $6 \times 10^{-12}$ points mean that net-removal of 1 tCO$_2$ with this layout brings combined environmental impacts equivalent to $6 \times 10^{-10}$% of the global annual total. To provide an environmental assessment of each energy system combination, we use the sum of all positive environmental impacts.

Results for different energy system configurations are shown in Fig. 7. Other than for costs and GWP, lowest combined environmental impacts are achieved for lower utilization rates

of 33% ("35–10"). This combination runs with only a small battery storage. For higher utilizations, however, the added environmental burden of increased battery capacity presents an issue. While the battery storage is not highly problematic from a cost and GWP perspective, using alternative energy storage options might be most suitable to reduce combined environmental impacts.

**Scenario analysis.** Some learning scenarios are assessed to gain insight into future potential of off-grid DACS. Including costs, GWP and combined environmental impacts, we conceive a dashboard which shows implications for each scenario. Results for the "100–300" system are set as 100% reference benchmark for costs, GWP and environmental impacts. Thus, it is easy to see what effect any scenario has on the three relevant metrics. A capex reduction of the DAC plant by 50% to $ 760 per tCO$_2$ and year (the recently announced specific capex of 1 MtCO$_2$ / year systems in the U.S.[16] is $ 600 per tCO$_2$ and year[16]), reduces net costs by 15.4%, see Fig. 8. Reduced adsorbent costs lower net costs by 10.3%. Interestingly, a related scenario, in which not adsorbent costs but adsorbent degradation is reduced by 50% brings more benefits to costs (−14.4%). This is explained by lower GWP and higher CRE of this scenario. Reducing the energy demand by 50% brings the highest benefit to costs (−23.9%). Relative benefits to GWP and combined environmental impacts are even larger. With lower energy consumption, the cost-efficient PV system has a power of 55 MW combined with a 160 MWh battery storage. Finally, if all improvements were to occur simultaneously, net

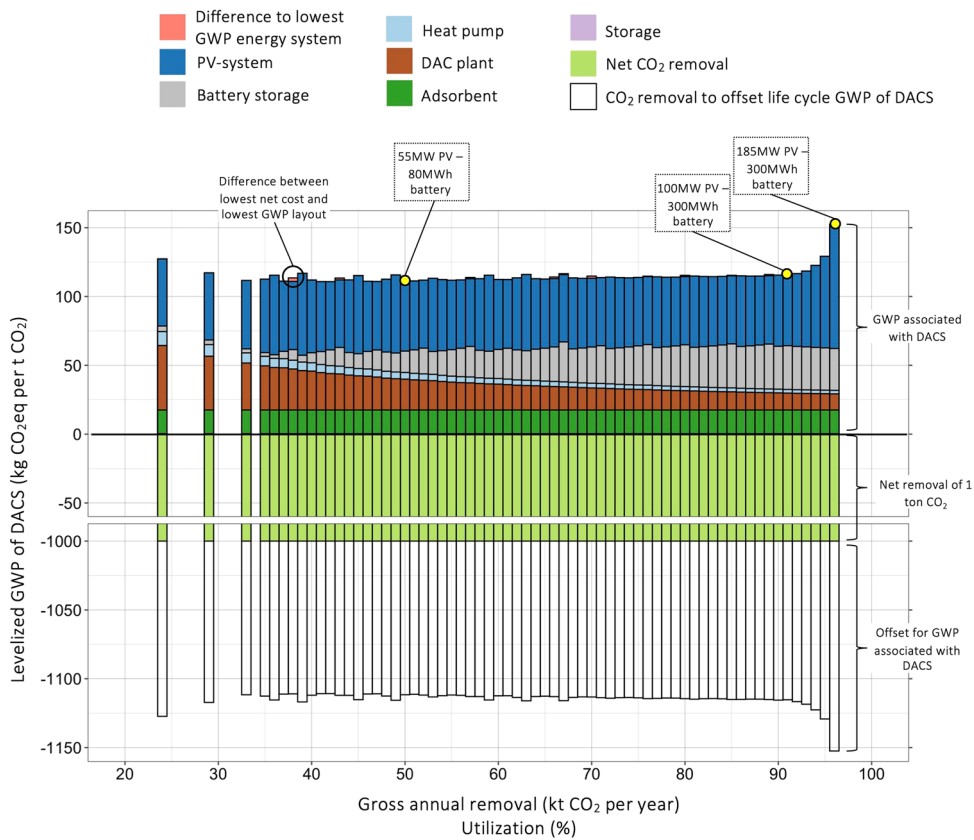

**Fig. 5 Global warming potential associated with net removal of 1 tCO₂ with off-grid DACS in Nevada (USA).** Production and consumption of DACS components are associated with emission of greenhouse gases. Thus, for the net removal of 1 tCO₂ more than 1 tCO₂ must be captured and removed to offset emissions associated with the process. Bars represent the global warming potential (GWP) for lowest net cost configurations for different utilization factors. Colours reflect the (positive) contribution of different components to GWP. Bars in light green represent the goal of net removing 1 tCO₂, with white bars accounting for the addition CO₂ capture and removal to offset (positive) greenhouse gas emissions of the capture and removal process.

costs of $ 369.5 per tCO₂ could be reached with a CRE of 93.6%. Including tax credits would bring costs down to $216.5. These costs are substantially lower than the $ 1200 per tCO₂ currently charged by commercial DACS operators[17]. With combined environmental impact reduced to $3.1 \times 10^{-12}$ points, the benefit of DACS for the impact category of climate change (which is $-3.8 \times 10^{-12}$ points) is larger than the sum of the environmental burden of the remaining 15 environmental impact categories. (Beware, however, of the uncertainty associated with normalization and weighing.)

If 20,000 systems of these optimized systems were installed in Nevada (or locations with similar climate and storage locations), this would represent a design capacity of 2 GtCO₂ per year. Operating at an average utilization of 91% with a 50 MW and 150 MWh battery each, a gross total of 1.82 GtCO₂ per year would be captured and removed. With a CRE of 93.6%, this is equivalent to the net-removal of 1.70 GtCO₂ per year. Normalized and weighted environmental impacts of these systems would account for 0.62% of global annual environmental impacts (on 2010 level), primarily for mineral resource scarcity. At the same time, global climate change impacts would be reduced by 3.2%. For each system, total upfront investment costs are $ 187.8 million (PV: 48.1, battery: 50.8, DAC plant: 76, heat pump: 12.9). Thus, the total upfront investments for 20,000 systems comes to $ 3,740 billion. Note that storage is treated as variable cost, so that infrastructure investments must occur by a third party.

A brief sensitivity analysis, conducted for a flat 30% increase or decrease in key technical parameters, shows interest rate, capex of DAC, and adsorbent costs as most relevant factors, see Fig. 9.

Therefore, providing DACS companies with access to funding with 7% interest rate rather than 10%, reduces costs by $ >90 per tCO₂, pointing to possible strategies for policy subsidies. Reducing the GWP of the adsorbent material or the battery storage has no effect on the gross costs of DACS. However, for net costs, reduced GWP associated with the production of the PV system or battery storage, has the potential to reduce net costs by $ 15. This might provide a basis for negotiation between manufacturers of adsorbent materials and DACS operators.

**Comparison to existing studies**. In ref. [12], DACs at 91% utilization (8000 full load hours) achieved lower LCOD$_{gross}$ ($ 222) than DACs at 46% utilization (4000 full load hours; $ 289). Cost contribution of the energy system to total LCOD were $ 115 and $ 74, respectively. In our calculation, cost contribution of the energy system with comparable utilization factors to LCOD$_{gross}$ are higher with $ 305 and $ 239, respectively. This gap is in part explained by higher heat and energy requirements per ton CO₂ captured for our reference with 3.33 MWh$_{heat}$ and 0.6 MWh$_{electricity}$. For a 100 ktCO₂ per year first of a kind (FOAK) DACS power by generic PV-electricity with a heat pump (but without accounting for intermittency) from the IEAGHG[13], LCOD$_{gross}$ were $ 607, with PV electricity accounting for $ 136. In our situation the PV system accounts for $ 116, which is lower than reported by ref. [13]. However, to mitigate intermittency, the battery storage adds another $ 159 to LCOD$_{gross}$, highlighting the necessity to include cost for electricity storage. For comparison, bioenergy with carbon capture and storage, another CO₂ removal technology, has been associated with costs between $ 80 - 319 per tCO₂ removed[32], which is lower

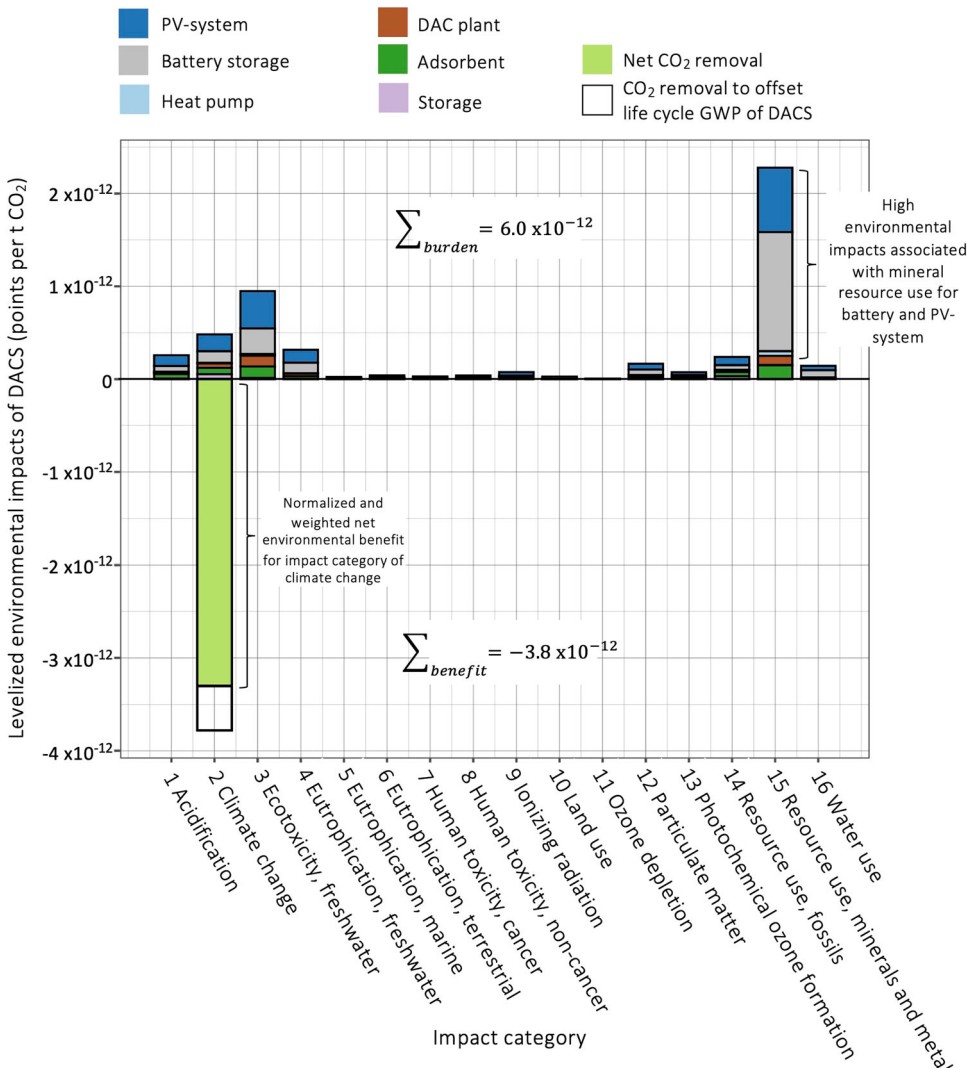

**Fig. 6 Environmental impacts associated with removal of 1 tCO$_2$ with off-grid DACS in Nevada (USA) equipped with a 100 MW photovoltaic system and 300 MWh battery storage.** Environmental impacts are shown for the 16 impact categories of the Environmental Footprint 3 life cycle assessment characterization model[47,48]. Results are normalized against annual global totals in each impact category and weighed with factors from the European Commission[31]. Colours of the bars reflect the contribution of different components to impacts. Negative environmental impacts (meaning benefits) are achieved for the impact category of climate change, while positive environmental impacts (meaning burdens) result in all other impact categories.

than our calculated DACS costs. However, BECCS does, by design, face food vs. fuel concerns and problems associated with biodiversity, which DACS does not[8,33].

GHG emissions of 100 ktCO$_2$ per year FOAK and nth of a kind (NOAK) DAC in ref. [13] are 123 and 41 kgCO$_2$eq per tCO$_2$ gross captured. In our reference scenario, contribution of the PV system is 51 kgCO$_2$eq, with another 32 kgCO$_2$eq from the battery storage. In a NOAK DAC, electricity of the PV-system accounts for 25 kgCO$_2$eq per 1 tCO$_2$ captured. In our optimized scenario, the energy system contributes 42.7 kgCO$_2$eq to total GWP (25.5 kgCO$_2$eq by PV system). A reason for the stronger decline in GHG emissions in ref. [13] was that the carbon footprint of PV electricity was assumed to decline from 51 to 25 kgCO$_2$eq per MWh, compounding the effect on reduced GWP emissions. We, on the other side, followed the example of refs. [20,27], and did not include reductions in background inventory data to limit complexity. Consequently, our environmental results (and projection about CRE) tend to come in on a more conservative side. Reductions in carbon intensity of the grid have effect only

on GWP of adsorbent for existing systems and on all components of future systems.

Specific energy consumption in our reference scenario and the 4 ktCO$_2$ per year DAC of ref. [20] is the same. In ref. [20], with a DAC powered by a generic PV-system (with Germany as location), energy accounts for 166 of the total 204 kgCO$_2$eq associated with gross removal of 1 tCO$_2$. Our reference scenario reports 85.4 kgCO$_2$eq associated with the energy system, which s lower. A key explanation for this difference is a reduced lifetime energy output of the PV-system for suboptimal solar-irradiation locations such as Germany.

In ref. [26], an autonomous 100 ktCO$_2$ per year off-grid DACS, powered by a PV-system and heat pump in Greece achieved a CRE of 86.4%, which is slightly lower than the CRE of 88.4% in our reference scenario. The energy system contributes 84.6 kgCO$_2$eq to total GWP of 1 tCO$_2$ removed (compared to 85.6 kgCO$_2$eq in our study). Of this, 28 kgCO$_2$eq are associated with the battery storage (compared to 31.8 kgCO$_2$eq in our work). In addition, the specific energy requirements of a 100 ktCO$_2$ per year

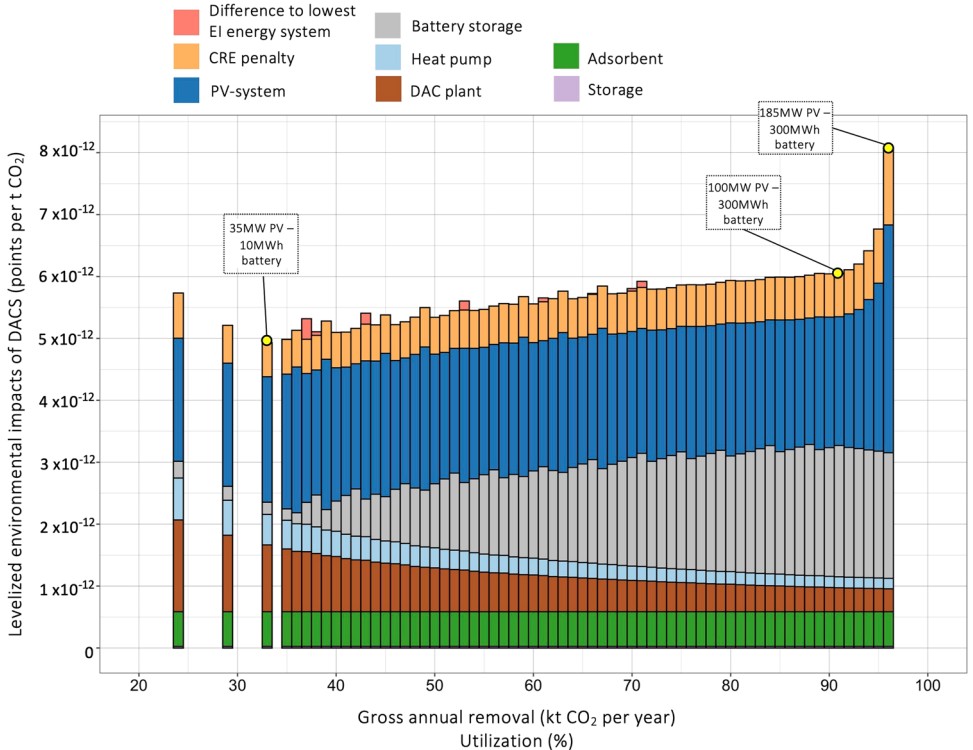

**Fig. 7 Normalized and weighed environmental impacts associated with removal of 1 tCO$_2$ with off-grid DACS in Nevada (USA).** Sum of all positive environmental impacts after normalization and weighting[31]. Bars represent the lowest net costs combinations for different utilization factors, with colours indicating the contribution of each component to total environmental impact scores. Yellow parts of bar indicate the difference between net and gross removal of 1 tCO$_2$, accounting for carbon removal efficiencies (CREs). Negative environmental impacts (benefits) for the impact category of climate change are $-3.8 \times 10^{-12}$ points per tCO$_2$ regardless of the utilization factor. Benefits of DACS in the climate change impact category are not included in this figure.

system in ref. [26] are the same as in the 100 ktCO$_2$ per year future setup in ref. [20] (that is, 1.5 MWh$_{heat}$ and 0.5 MWh$_{electricity}$). As such, energy consumption is close to our optimized scenario. With lower energy consumption, the contribution of the energy system is 42.7 kgCO$_2$eq in our work, of which 16 kgCO$_2$eq come from the battery storage. This is lower than in ref. [26]. A battery capacity of 221 MWh was used by ref. [26], combined with a PV-system of 67–132 MW (the exact size depending on the location). With this energy system, no reduction in full load hours of the DAC was discussed. The optimal energy system combination in our optimized scenario is a 50 MW PV system combined with a 150 MWh battery storage.

## Conclusion

Consensus is growing that upscaling of DACS is an integral part of net-zero strategies[4,10,13,15,34]. High CREs are achieved if renewable energy systems are used to provide the required energy. To date, intermittency problems of renewable energy systems such as wind and photovoltaic have not been comprehensively addressed. Off-grid DACSs have the substantial benefit that companies operating the systems do not have to wait until the location-specific grid electricity reaches a low CO$_2$ intensity, and can start right away, which is crucial to reach scale.

The present work provided an approach for optimizing the design of an off-grid low-temperature, solid sorbent DACS powered by PV system, battery storage and heat pump based on cost and environmental impact assessments. LT solid sorbent DACS were selected because this system design is expected to better deal with on-off operating mode than high-temperature liquid solvent systems[13]. For a design capacity of 100,000 tCO$_2$ per year we built a model to estimate the utilization factor of the

DAC plant for PV systems with 5–200 MW and LIBs with 0 to 300 MWh capacity. For each combination, costs and environmental impacts were calculated and least-cost, least-GWP or least-environmental impact setups identified.

Some limitations should be highlighted. First, our focus was on PV powered systems. Future work should expand the model to optimize the DACS including both wind and PV power for electricity generation. Another limitation is that only lithium-ion batteries were included for energy storage. Future work could include a heat storage system in addition to a battery. Also, our work did not include high-temperature liquid solvent DACS, the other major class of DAC technologies. Off-grid operation of high-temperature DACS might present additional challenges. Nevertheless, future work could attempt to optimize such an off-grid system and compare costs and environmental impacts to the one presented here.

Our integrated approach has shown that for PV and battery powered DACSs, selecting the optimal energy system combination based on minimal net costs for 1 tCO$_2$ removed also brings overall environmental impacts within a 20% range to combinations chosen based on minimum environmental impacts. Hence, our work indicates that selecting the energy system based on lowest net-costs is also a reasonable choice from a life cycle assessment perspective. For an American use-case, with average per capita GHG emissions of 17.5 tCO$_2$eq per year (ref. [35]), annual costs for people willing to compensate their entire emissions with an off-grid DACS in the reference scenario would be $ 15,300 per year. In the optimized scenario, costs are around $ 6450 per year and $ 3,780 with U.S. 45Q tax credits. Compared to an average GDP per capita of $ 70,000 in 2021 (ref. [36]) this suggests that compensation of GHG emissions alone through DACSs would require between 22% and 5% of GDP.

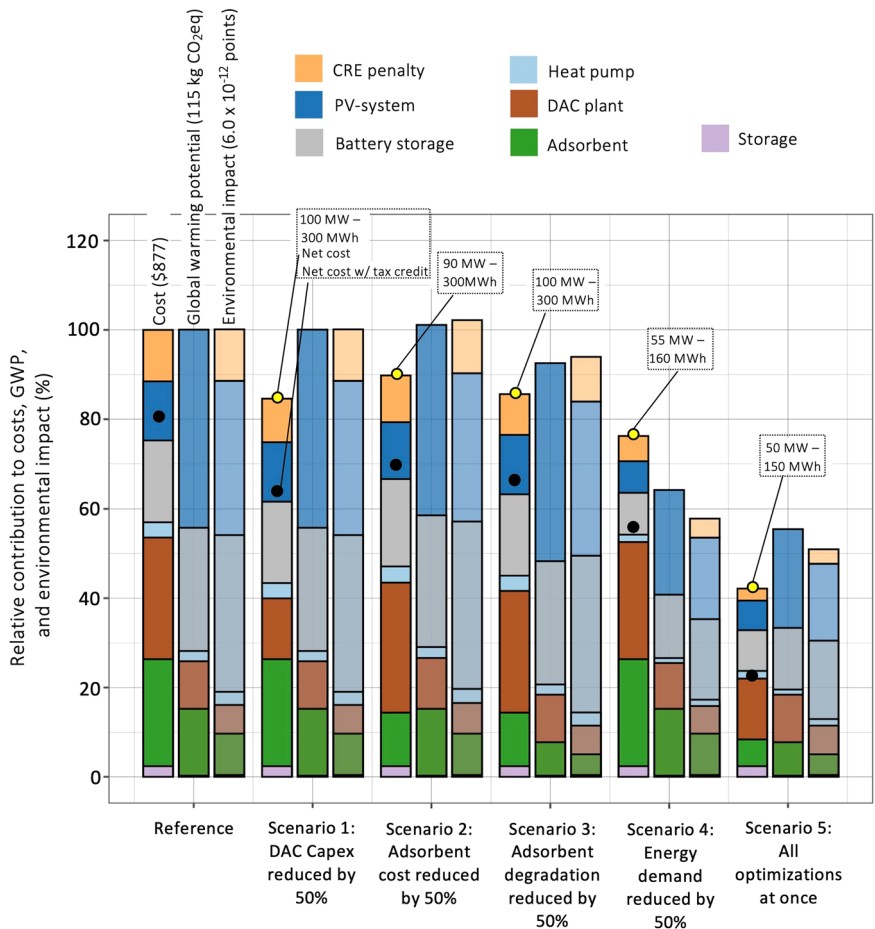

**Fig. 8 Economic-environmental dashboard for strategic learning scenarios of off-grid DACS.** Results for the cost optimal DACS layout in Nevada (USA) with a 100 MW PV-system and 300 MWh battery represent the 100% benchmark for costs, GWP, and combined environmental impacts. Black dots show costs if the 45Q tax credit is included. Colours used in bars represent the contribution of components to total results. Yellow parts of bar indicate the difference between net and gross removal of 1 $tCO_2$, accounting for carbon removal efficiencies (CREs).

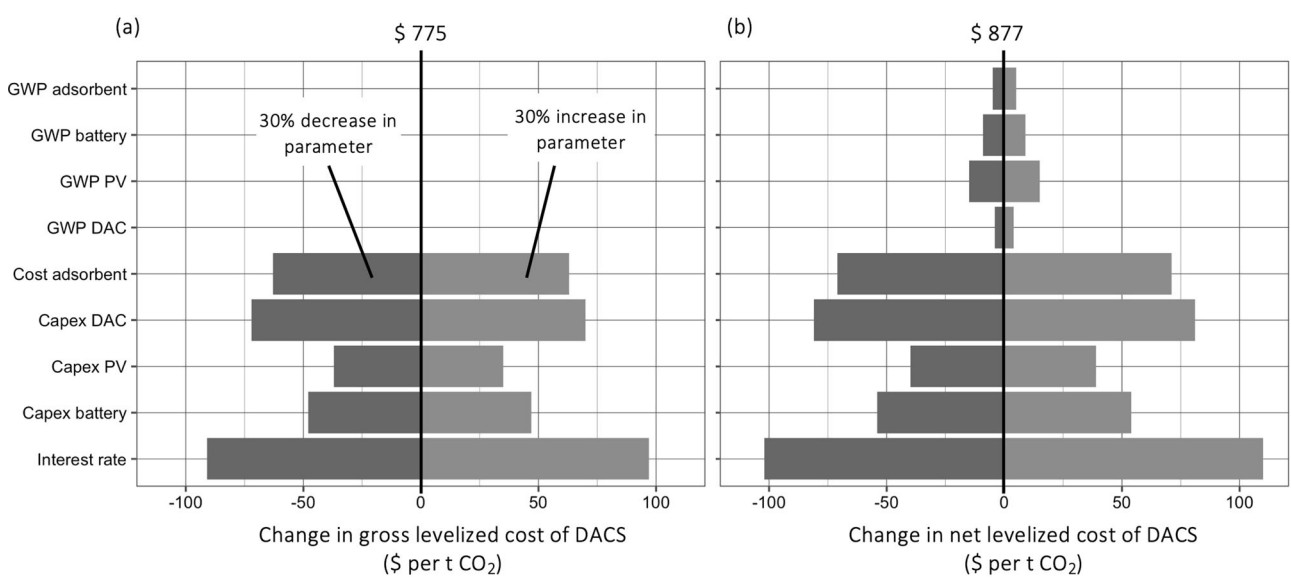

**Fig. 9 Sensitivity analysis on levelized costs for gross and net removal of 1 $tCO_2$ with off-grid DACS in Nevada (USA) equipped with a 100 MW PV-system and 300 MWh battery storage.** Impact of 30% decrease and 30% increase of parameters on levelized costs for (**a**) gross removal and (**b**) net removal of 1 $tCO_2$ from the atmosphere.

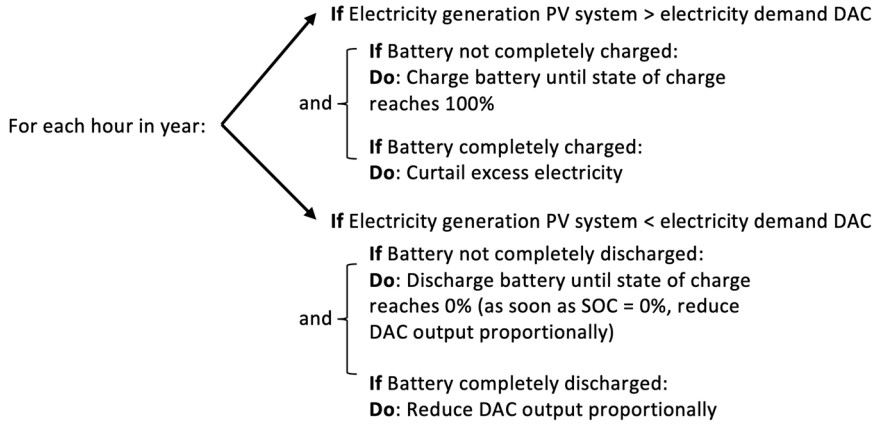

**Fig. 10 Interaction between PV-electricity generation, battery storage and energy consumption by DACS system.** For every hour in a representative year (2020) the model structure allows to calculate the hourly $CO_2$ capture rate of the DACS. Summing up results for 1 year gives estimates for the annual $CO_2$ removal and utilization factor.

## Methods

Our methodology consists of different parts to answer the research questions posed in the introduction. First, a model is built to estimate the utilization factor (full load hours) of the proposed off-grid DACS for various combinations of photovoltaic power, capacity of the battery storage and location. Second, we provide the structure for LCA and cost assessment, using a shared system boundary of the DACS (shown in Fig. 1). Data from the utilization model is then used to calculate total costs and environmental impacts of the DACS for different combinations. Further, our approach allows to optimize the energy system to achieve lowest costs per 1 tCO$_2$ gross removed or lowest cost for 1 tCO$_2$ net removed, thereby including LCA GWP for the same layout to calculate CRE as efficiently as possible. In addition, energy setups with lowest contribution to environmental impact categories can be identified and trade-offs discussed.

For off-grid, PV powered systems, locations with high solar irradiation and substantial $CO_2$ underground storage potential are of interest. The U.S. Geological Survey[37] and ref. [38] show the geographical distribution of suitable saline aquifer basins for underground carbon dioxide mineralization. To keep transportation distances of captured $CO_2$ to a minimum, the proposed DACS system is placed on top of a saline aquifer basin to reduce transport distances. With solar irradiation data from ref. [39], we select Las Vegas (Nevada, USA), Swakopmund (Namibia), and Munich (Germany) as location for the analysis with suitable geological storage potential nearby[38]. The underground storage of $CO_2$ must be durable, that is, without significant leakage, to effectively reduce atmospheric $CO_2$ levels and mitigate risks. Work on the security of underground $CO_2$ storage predicts that the cumulative $CO_2$ leakage will be 1.8–25% for a timespan of 10,000 years[30]. This is equivalent to an annual leakage rate of 0.00018–0.0025%, which is very small compared to GHG emissions associated with the energy used for capture and storage of $CO_2$.

**Utilization model**. An off-grid DACS, powered by intermitted, renewable energy sources, is subject to fluctuations in electricity generation. For example, the energy generation with PV-systems depends strongly on the solar irradiation. If the sun does not shine, the PV-system does not generate electricity. Thus, a DACS system directly linked to the PV-system would have to stop or reduce operation whenever solar irradiation is low. Energy storage mediums, such as battery storage systems, can store electric energy for later use, thereby increasing the

availability of the DACS system. However, adding battery capacity involves costs and environmental impacts. We conceive a basic structure of a model to estimate the annual operating hours and captured $CO_2$ for numerous PV-systems, battery storage capacities, and locations.

Hourly data for local solar irradiation as well as power output of the PV system is obtained via the Photovoltaic Geographic Information System (PVGIS) from the European Commission[39]. Reference year is 2020. Multiplying power output for a generic 1Wp system with the power chosen for the energy system leads to hourly electricity generation of the PV system.

On the consumption side, hourly demand for direct electricity of fans and compression unit are calculated based on the design capacity of the DAC plant and specific electricity requirements (see also Supplementary Note 1). Similarly, heat requirement of the DAC plant, for regeneration of saturated sorbent material, is transformed to electricity demand using a high-temperature heat pump. For heat pumps, the coefficient of performance (COP) indicates how much thermal energy is gained for each unit of electricity input (kWh$_{thermal}$ per kWh$_{electricity}$). Regeneration of solid sorbent material requires temperatures of 100 °C. This level can be reached with state-of-the-art industrial, high temperature heat pumps[40]. The COP decreases for lower temperatures of the heat source. As the air temperature differs for each location, so does the COP. Using average air temperatures, we estimate COPs of 2.6 (Las Vegas, Nevada), 2.5 (Swakopmund, Namibia), and 2.3 (Munich, Germany) based on ref. [40]. We assume that energy demand for a DAC plant operating at full capacity is constant. Thus, it is possible to calculate the hourly energy demand of the DAC. Matching of electricity supply and combined electricity demand takes the following form shown in Fig. 10.

Using this approach, utilization of the DAC plant is calculated for different combinations of the energy system. The design capacity in the present work was set to 100,000 tCO$_2$ per year, similar to the "small" system of the ref. [13] and the reference system size in ref. [26]. Ranges for the PV system are set between 0 and 200 MW and battery capacity between 0 and 300 MWh, with 5 MW steps for PV and 10 MWh steps for the battery. Combinations are denoted as "P-C" ("Power-Capacity") for power of PV system in MW and capacity of the battery in MWh.

**Cost evaluation**. Previous work has established levelized cost concepts to evaluate economic aspects of DAC systems, see for

example, Eq. 2 (ref. [12]) and Eq. 3, (ref. [13]).

$$LCOD_{gross} = \frac{Capex_{DAC} * crf + Opex_{fix}}{Output_{CO_2,captured}}$$
$$+ Opex_{var} + Heat * Levelized\ cost\ of\ heat\ (LCOH)$$
$$+ Electricity * Levelized\ cost\ of\ electricity\ (LCOE)$$
(2)

$$LCOD_{gross} = \frac{Capex_{DAC} * crf + Opex_{fix} + Opex_{var} * annual\ CO_{2,captured}}{annual\ CO_{2,captured}}$$
(3)

Combined with the CRE, LCOD can be transferred to levelized cost for net capture/removal of 1 tCO$_2$ from the atmosphere (see Eq. 4, refs. [13,24]).

$$LCOD_{net} = \frac{LCOD_{gross}}{CRE}$$
(4)

To estimate costs, we build on this levelized cost approach. The present work distinguishes between annualized costs, and levelized costs per ton CO$_2$ gross and net removed. Dividing levelized costs for gross removal of 1 tCO$_2$ by the carbon removal efficiency (calculated with Eq. 1 using data from the simultaneously conducted LCA), leads to levelized cost for net removal of 1 ton CO$_2$ (Eq. 4).

$$Annualized\ cost\ of\ DACS\ (ACOD) = Capex_{DAC} * crf + Capex_{PV} * crf$$
$$+ Capex_{Battery} * crf + Capex_{HP} * crf$$
$$+ (Opex_{variable,Adsorbent}$$
$$+ Opex_{variable,Trans\&Storage})$$
$$* annual\ CO_2\ gross\ captured$$
(5)

$$LCOD_{gross} = \frac{ACOD}{annual\ CO_2\ gross\ captured}$$
(6)

$$Capital\ recovery\ factor\ (crf) = \frac{i * (1 + i)^{Lifetime}}{(1 + i)^{Lifetime} - 1}$$
(7)

Equations below specify the calculation of fixed costs (Eqs. 8–11) and variable costs (Eqs. 12,13). We treat costs for the energy infrastructure as fixed, as investment into the energy system is part of the proposed off-grid design.

$$Capex_{DAC}[\$] = Design\ capacity\left[\frac{t}{year}\right] * spec.Capex_{DAC}\left[\$ * \frac{year}{t}\right]$$
(8)

$$Capex_{PV}[\$] = Power\ PV\left[MWp\right] * spec.Capex_{PV}\left[\frac{\$}{MWp}\right]$$
(9)

$$Capex_{Battery}[\$] = Capacity\ Battery\left[MWh\right] * spec.Capex_{Battery}\left[\frac{\$}{MWh}\right]$$
(10)

$$Capex_{HP}[\$] = Power\ HeatPump\left[MWth\right] * spec.Capex_{HP}\left[\frac{\$}{MWth}\right]$$
(11)

Variable costs

$$Opex_{variable,Adsorbent}[\$/t] = Sorbent\ consumption\ [kg/t]$$
$$* spec.Cost_{Adsorbent}\left[\frac{\$}{kg}\right]$$
(12)

$$Opex_{variable,Trans\&Storage}[\$/t] = spec.Cost_{Transport}\left[\frac{\$}{t * km}\right]$$
$$* Transport\ Distance\ [km] + spec.Cost_{Storage}\left[\frac{\$}{t}\right]$$
(13)

Labour costs are not included since DACS operation is a highly automated process, and previous work has shown that labour costs account for <1% of total costs[24]. For any U.S. location, the 45Q credit for carbon dioxide removal is of high relevance[41]. As stated therein, DACS systems are eligible for a $ 180 per tCO$_2$ credit (in the form of tax credit or direct pay)[42]. The duration of the credit is 12 years, which is almost the same as the 12.5 years lifetime assumed for the DAC plant in Supplementary Table 3, leading to an adjusted tax credit to $ 172 per tCO$_2$ over the complete lifetime of the DACS system. Also, the 45Y credit for clean electricity production subsidizes electricity generated by renewable energy systems by up to $ 26 per MWh for 10 years. Due to the off-grid design, however, no electricity is sold to the grid, arguably reducing the relevance of 45Y credits for the present design.

**Life cycle assessment.** Generally, LCAs aim to quantify the environmental impacts associated with a product or service (permanent removal of 1 tCO$_2$ from the atmosphere can be thought of as a service). The ISO 14040/44 provides a structure on how LCAs should best be conducted[43,44]. Defining a functional unit (FU), system boundaries and impact categories are key features[43]. In the present work cost evaluations are accompanied by a LCA in line with ISO 14040/44. Getting GHG emissions associated with CO$_2$ removal is necessary to calculate costs for net removal of CO$_2$.

In addition, insight into environmental impact categories apart from climate change help to comprehensively assess environmental challenges and benefits of DACSs. Thus, the goal of the present LCA is to calculate environmental impacts for an off-grid low-temperature, solid-sorbent DACS powered by a heat pump and various combinations of PV system and LIB storage. A state-of-the-art LIB with NMC$_{811}$ as cathode material is selected due to its competitive price[45] and large-scale industrial production[46]. Two functional units (FUs) are assessed in line with the overall goal of the present paper. The first FU is "gross capture and removal of 1 ton CO$_2$" (FU1) the second FU is "net capture and removal of 1 ton CO$_2$" (FU2). EF 3.0 with 16 impact categories is used as characterisation model[47,48]. Life cycle inventory data is gathered from literature sources which had access to primary LCI data for solid-sorbent DAC processes[20,26,27] and Ecoinvent 3.8 (ref. [49]). Normalization and weighting factors for environmental impact categories are obtained from the Euorpean Commission[31]. Following a comparable approach to cost calculations we calculate annualized environmental impacts (EI) of DACS (Eq. 14) and levelized environmental impacts of DACS for 1 ton gross removal of CO$_2$ (FU1) and 1 ton net removal of CO$_2$ (FU2) from the atmosphere (Eqs. 15,16). We assume that all

captured $CO_2$ is removed, with an annual leakage rate of 0.0025%.

$$AEOD = EI_{DAC} * \frac{1}{lifetime} + EI_{PV} * \frac{1}{lifetime} + EI_{Battery} * \frac{1}{lifetime}$$
$$+ EI_{HP} * \frac{1}{lifetime} + (EI_{variable,Adsorbent} + EI_{variable,Trans \& Storage})$$
$$* \text{ annual } CO_2 \text{ gross captured}$$

(14)

$$LEOD_{gross} = \frac{AEOD}{annual \ CO_2 \ gross \ captured}$$

(15)

$$LEOD_{net} = \frac{LEOD_{gross}}{CRE}$$

(16)

**Integrating utilization model, cost model and LCA**. The utilization model allows to estimate the full load hours of the DAC plant for given energy consumption parameters. As outlined before, power of the PV system and capacity of battery are required input parameters for both LCA and cost model. Using the calculated full load hours (corresponding with annual gross capture and removal of $CO_2$) for each "P-C" combination as input allows us to transparently provide environmental impacts and costs for DACSs with different energy system combinations.

Key data sources for technology, cost and environmental parameters are gathered through a review of recent literature (see Supplementary Tables 1 and 2). To reflect 2023 levels, cost data for consumables is adjusted with the consumer price index (CPI)[50] and for investments with the chemical engineering plant cost index (CEPCI)[51]. Thus, data in Supplementary Tables 3 and 4 represent a reference scenario for a state-of-the-art plant based on current LT solid sorbent technology.

**Developing scenarios based on reference situation**. Prior studies have accounted for improvements in technology through learning. Here we look at several potential learning scenarios (see Supplementary Table 5) to evaluate the combined effect on cost and environmental impacts as well as the optimal combination for the energy system.

## Data availability

Data supporting the results of the present study are presented in the Supplementary Information. LCI data for different technology scenarios as well as LCI data sources are provided in the Supplementary Note 2. Data sources for technology and cost scenarios are given in the paper.

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

## Author contributions

MG conceived the idea, designed the research, and performed the analysis. JL guided and supervised the project. MG and JL wrote and revised the manuscript.

## Funding

## Competing interests

The authors declare no competing interests.
