## [Peer review file · Communications Engineering]

Co-assessment of costs and environmental impacts for off-grid direct air carbon capture and storage systemsThis manuscript has been previously reviewed at another Nature Portfolio journal. This document only contains reviewer comments and rebuttal letters for versions considered at Communications Engineering.

Reviewer #1 (Remarks to the Author):

In the manuscript "Co-assessment of costs and environmental impacts for off grid direct air carbon capture and storage (DACS) systems", the authors performed TEA and LCA to quantify the costs and environmental impacts of performing DAC under different scenarios in Arizona. Overall, this work is well-written and well-structured, and the authors provide sufficient details to support their statements in most cases. I think this work would be a nice contribution to the existing literature, and I would recommend this work to be published after revisions. Below are my comments that I hope the authors would find useful.

1. Page 2, line 45-46. Is TRL of 9 for small commercial scale or for full-scale? This is minor but may worth clarifying.
2. Page 3, line 90-91. I do agree that LT systems put less requirements on water supply. But in my opinion, this is because in LT systems the cooling duty is much lower as compared to in HT systems. Could the authors verify which one (reduced cooling or water recovery) is the main contributor to the reduction in water demand?
3. This is a general comment. Since the reference location is Arizona, I strongly recommend that the authors add tax credits from the 2022 Inflation Reduction Act in the LCOD calculations. According to 2022 IRA, DAC could qualify a 12-year 45Q credits at \$130/tonne CO₂ or \$180/tonne CO₂, depending on whether CO₂ is for utilization or permanent storage (the case in this study) (<https://www.iea.org/policies/16255-inflation-reduction-act-2022-sec-13104-extension-and-modification-of-credit-for-carbon-oxide-sequestration>). Additionally, the solar/wind generator may additionally qualify 45Y credits at \$26/MWh for zero-carbon electricity generation, which may further reduce the LCOD. It should be noted that the duration of these tax credits may be shorter than the project's lifetime, so the credits need to be derated, as suggested by this work (<https://arxiv.org/abs/2305.00946>).
4. This is also a general comment. The US DOE has announced two major DAC projects (<https://www.energy.gov/articles/biden-harris-administration-announces-12-billion-nations-first-direct-air-capture>), one in Texas and one in Louisiana. Both projects would capture 1 million metric tons of CO₂ every year. The one in Louisiana uses LT systems. The total cost announced is \$1.2 billion, so the average CAPEX would be \$600/(tonne CO₂/year). This CAPEX is lower than the value from source 13. While I don't think the authors need to modify anything in the existing calculations, I suggest the authors briefly discuss about this timely piece of information.
5. This is another general comment. How does the cost estimate of DAC in this work compare to other technologies that provide similar negative emission benefits (e.g., BECCS)?
6. Page 9, line 200-201. Would the GHG emission decrease as other sectors get decarbonized?
7. Page 19, line 434. Have the authors considered other energy storage systems (e.g., thermal storage in bricks) that are potentially cheaper than LIB? Is the selection of LIB based on its high TRL or low cost?
8. Page 20, line 481. I noticed that the authors also considered other locations for DAC, does this COP apply in all regions (since the temperature is different)?
9. Page 22, Table 1. Could the authors elaborate on how this 100 km is decided? Does Arizona have saline aquifers (or equivalent) for CO₂ storage?
10. Page 22, Table 2. Could the authors please add the dollar year to the table? Have these cost items been adjusted using CEPCI (for equipment) and CPI (for consumables). Doing the CEPCI and CPI corrections are rather important. Also, would there be labor cost?

Reviewer #2 (Remarks to the Author):

The manuscript presents an assessment of the cost and environmental impacts of an off-grid DACS (Direct Air Capture and Storage) system using solid sorbents. The work relies on literature data. It includes scenario analysis of future learning pathways and sensitivity analysis to pinpoint key parameters affecting the results. The conclusions are robust and supported by the analysis presented

Overall, I believe the manuscript is interesting, original (though primarily based on literature data) and thus worth publishing. I have also noticed that the Authors have addressed (to some extent) most of the comments of the previous round of review, which significantly improved the quality of the manuscript.

Below are my main comments:

- I am not clear whether the analysis considers leakages of CO₂ from transportation and permanent sequestration. As far as I am aware, literature estimates for both are low; but it is, in my opinion, worth at least discussing this, if not including both parameters in the sensitivity analysis.
- The charts are nice, but not always easily understandable. For example, it is not clear to me what the grey dots (exponential curve) in Figure 3 represent, and how they differ from those vertical. I strongly suggest to include the full description of all components of the charts in the caption, and the symbol in the legend. Note that the same comment also applies to the other figures, e.g. Figure 3 and the black dots, which should all be checked and amended accordingly.
- The Authors should better describe the trade-offs associated with DACS, that is it has "negative" climate change impact but "positive" impacts in all other categories.
- The weighting factors used in the LCA study to obtain a single score describing the overall environmental performance are inherently value-based. This must be highlighted, particularly that different weighting factors may yield different results.
- The concept of the "utilization" of the DAC plant is particularly important, but it is in my opinion not introduced with sufficient clarity in the manuscript. A reader who is not familiar with it may struggle understanding the results of the manuscript.
- The sensitivity analysis is interesting, but I am not clear how this has been conducted. Have the Authors used plausible ranges for the parameters (which may differ for different parameters)? Or have they assumed the same range of variation? The interpretation of the results of the sensitivity analysis are heavily dependent on this.

Dear Reviewers,

thank you very much for reviewing and commenting on our manuscript. Addressing your comments and concerns has helped us to improve the quality of the manuscript and increase its appeal to an audience with different backgrounds.

Please find below a point-by-point response to your comments.

Reviewer #1

Comment 1.1

In the manuscript “Co-assessment of costs and environmental impacts for off grid direct air carbon capture and storage (DACs) systems”, the authors performed TEA and LCA to quantify the costs and environmental impacts of performing DAC under different scenarios in Arizona. Overall, this work is well-written and well-structured, and the authors provide sufficient details to support their statements in most cases. I think this work would be a nice contribution to the existing literature, and I would recommend this work to be published after revisions. Below are my comments that I hope the authors would find useful.

Answer 1.1

We appreciate your feedback and favorable comments about the structure of our work.

Comment 1.2

Page 2, line 45-46. Is TRL of 9 for small commercial scale or for full-scale? This is minor but may worth clarifying.

Answer 1.2

Thank you for highlighting this inconsistency. We have double-checked this point with the references and now state that:

“Technological readiness levels for both designs are between 6 (pilot plant) and 8 (small commercial scale)^{8,9}.”

Comment 1.3

Page 3, line 90-91. I do agree that LT systems put less requirements on water supply. But in my opinion, this is because in LT systems the cooling duty is much lower as compared to in HT systems. Could the authors verify which one (reduced cooling or water recovery) is the main contributor to the reduction in water demand?

Answer 1.3

We agree that the statement in the original manuscript was not as concise as possible. We looked into the technical descriptions provided by the IEA, which is in line with your suggestion. Thus we emphasize that lower water requirements of low-temperature systems is primarily due to lower cooling requirements, with co-extraction of atmospheric water as a potential benefit. Thus, the manuscript now states:

“In addition, LT systems put less requirements on water supply since cooling requirements are lower than for high-temperature systems, increasing the number of suitable locations.¹⁰ Potentially, LT systems could even provide benefits as a local water source since water is co-extracted from the atmosphere alongside CO₂¹⁰.”

Comment 1.4

This is a general comment. Since the reference location is Arizona, I strongly recommend that the authors add tax credits from the 2022 Inflation Reduction Act in the LCOD calculations. According to 2022 IRA, DAC could qualify a 12-year 45Q credits at \$130/tonne CO₂ or \$180/tonne CO₂, depending on whether CO₂ is for utilization or permanent storage (the case in this study) (<https://www.iea.org/policies/16255-inflation-reduction-act-2022-sec-13104-extension-and-modification-of-credit-for-carbon-oxide-sequestration>). Additionally, the solar/wind generator may additionally qualify 45Y credits at \$26/MWh for zero-carbon electricity generation, which may further reduce the LCOD. It should be noted that the duration of these tax credits may be shorter than the project’s lifetime, so the credits need to be derated, as suggested by this work (<https://arxiv.org/abs/2305.00946>).

Answer 1.4

Thank you for bringing up the tax credits currently provided by the U.S. government. This comment has had profound impact on our results. Throughout the manuscript, we have included a discussion of tax credits and have adapted our Figures accordingly.

We have added a paragraph in the Method section:

“For any U.S. location, the 45Q credit for carbon dioxide removal is of high relevance⁴³. As stated therein, DACS systems are eligible for a \$ 180 per tCO₂ credit (in the form of tax credit or direct pay)⁴⁴. The duration of the credit is 12 years, which is almost the same as the 12.5 years lifetime assumed for the DAC plant in Table 1, leading to an adjusted tax credit to \$ 172 per tCO₂ over the complete lifetime of the DACS system. Also, the 45Y credit for clean electricity production subsidizes electricity generated by renewable energy systems by up to \$ 26 per MWh for 10 years. Due to the off-grid design, however, no electricity is sold to the grid, arguably reducing the relevance of 45Y credits for the present design.”

And adapted the result section and figures:

“Recently, U.S. legislation has increased the tax credit for carbon capture with DAC systems to \$ 180 per tCO₂ removed. Fig. 4 includes this tax credit, with a small discount to match the longer lifetime of the DAC plant compared to the duration of the 45Q tax credit. Since credits are supposedly given for the net removal of CO₂, DACS systems with higher carbon removal efficiencies get slightly higher credits. Overall, credits lead to \$ 724.2 as lowest costs for net removal of 1 tCO₂ (“100-300”).”

Fig. 4 Levelized costs for gross and net removal of 1 tCO₂ with off-grid DACs in Nevada (USA). Bars represent the lowest (net) cost configurations of the PV-system and battery storage for different utilization factors. Colors of the bars reflect the contribution of each component (depreciation of battery, etc.) to total costs for the permanent removal of 1 tCO₂. Yellow parts of bar highlight the difference between costs for gross and net removal of 1 tCO₂. Costs for net removal of 1 tCO₂ include the carbon removal efficiency (which is <100%) for each configuration. Black dots represent the costs for net removal of 1 tCO₂ if the 45Q tax credit is subtracted from net costs.

Fig. 8 Economic-environmental dashboard for strategic learning scenarios of off-grid DACS. Results for the cost optimal DACS layout in Nevada (USA) with a 100 MW PV-system and 300 MWh battery represent the 100% benchmark for costs, GWP, and combined environmental impacts. Black dots show costs if the 45Q tax credit is included. Colors used in bars represent the contribution of components to total results. Yellow parts of bar indicate the difference between net and gross removal of 1 tCO₂, accounting for CREs.

“Finally, if all improvements were to occur simultaneously, net costs of \$ 369.5 per tCO₂ could be reached with a CRE of 93.6%. Including tax credits would bring costs down to \$216.5. These costs are substantially lower than the \$ 1,200 per tCO₂ currently charged by commercial DACS operators¹⁷.”

Further, we now include the 45Q tax credit in our conclusion:

“In the optimized scenario, costs are around \$ 6,450 per year and \$ 3,780 with U.S. 45Q tax credits. Compared to an average GDP per capita of \$ 70,000 in 2021 (ref.³⁸) this suggests that compensation of GHG emissions alone through DACSs would require between 22% and 5% of GDP.”

Comment 1.5

This is also a general comment. The US DOE has announced two major DAC projects (<https://www.energy.gov/articles/biden-harris-administration-announces-12-billion-nations-first-direct-air-capture>), one in Texas and one in Louisiana. Both projects would capture 1 million metric tons of CO₂ every year. The one in Louisiana uses LT systems. The total cost announced is \$1.2 billion, so the average CAPEX would be \$600/(tonne CO₂/year). This CAPEX is lower than the value from source 13. While I don't think the authors need to modify anything in the existing calculations, I suggest the authors briefly discuss about this timely piece of information.

Answer 1.5

Thank you for this reference suggestion. We have included this recent announcement in the introduction:

“Further, in August 2023, the U.S. Department of Energy announced plans to spend \$ 1.2 billion on the construction of DACS systems with a total capacity of 2 MtCO₂ / year¹⁶.”

to underline the fact that interest in DACS is growing. Further, we use the US DOE data to put the learning scenarios evaluated in our work into perspective:

“A capex reduction of the DAC plant by 50 % to \$ 760 per tCO₂ and year (the recently announced specific capex of 1 MtCO₂ / year systems in the U.S.¹⁶ is \$ 600 per tCO₂ and year¹⁶), reduces net costs by 15.4%, see Fig. 8.”

Comment 1.6

This is another general comment. How does the cost estimate of DAC in this work compare to other technologies that provide similar negative emission benefits (e.g., BECCS)?

Answer 1.6

We agree that putting the cost estimates of our work into perspective with other negative emission technologies is interesting. Thus, we have provided a brief perspective about BECCS within the comparison to existing studies:

“[...]In our situation the PV system accounts for \$ 116, which is lower than reported by IEAGHG¹³. However, to mitigate intermittency, the battery storage adds another \$ 159 to LCO_D_{gross}, highlighting the necessity to include cost for electricity storage. For comparison, bioenergy with carbon capture and storage, another CO₂ removal technology, has been associated with costs between \$ 80 - 319 per tCO₂ removed³⁴, which is lower than our calculated DACS costs. However, BECCS does, by design, face food vs. fuel concerns and problems associated with biodiversity, which DACS does not^{8,35}.”

Comment 1.7

Page 9, line 200-201. Would the GHG emission decrease as other sectors get decarbonized?

Answer 1.7

This is a good point. Since the PV-system, battery storage and DAC plant is bought today, any future reduction in GHG emissions of the world would not impact these components. However, since the adsorbent material is replaced continuously as it degrades, this would reduce the environmental impacts associated with adsorbent material use and increase the carbon removal efficiency. We state that:

“[...] Consequently, our environmental results (and projection about CRE) tend to come in on a more conservative side. Reductions in carbon intensity of the grid have effect only on GWP of adsorbent for existing systems and on all components of future systems.”

This, combined with our revised sensitivity analysis, provides a perspective on the potential of decarbonizing the industries which produce adsorbent material, PV-systems, DAC plants, and battery storages:

“Reducing the GWP of the adsorbent material or the battery storage has no effect on the gross costs of DACS. However, for net costs, reduced GWP associated with the production of the PV system or battery storage, has the potential to reduce net costs by \$ 15. This might provide a basis for negotiation between manufacturers of adsorbent materials and DACS operators.”

Fig. 9 Sensitivity analysis on levelized costs for gross and net removal of 1 tCO₂ with off-grid DACS in Nevada (USA) equipped with a 100 MW PV-system and 300 MWh battery storage. Impact of 30% decrease (green) and 30% increase (red) of parameters on levelized costs for gross removal (left) and net removal (right) of 1 tCO₂ from the atmosphere.

Comment 1.8

Page 19, line 434. Have the authors considered other energy storage systems (e.g., thermal storage in bricks) that are potentially cheaper than LIB? Is the selection of LIB based on its high TRL or low cost?

Answer 1.8

The concerns you raise here are completely valid. We have tried our best to find a compromise between addressing your concern and being transparent about limitations of our work. We provide an explanation for the selection of the lithium-ion battery in the method section, where we write that:

“A state-of-the-art LIB with NMC₈₁₁ as cathode material is selected due to its competitive price⁴⁷ and large-scale industrial production⁴⁸.”

With regard to other energy storage technologies, we state early on that balancing costs and benefits we have, for the present work, opted for a focussed approach on one energy storage technology:

“[...] Using a battery will also fill brief intermittency-gaps during the day to allow for a smoother operation of the DAC. Heat storage systems could also work but are excluded from the present analysis to limit complexity.”

And still include this as an area for future work:

“[...] Another limitation is that only lithium-ion batteries were included for energy storage. Future work could include a heat storage system in addition to a battery”.

Comment 1.9

Page 20, line 481. I noticed that the authors also considered other locations for DAC, does this COP apply in all regions (since the temperature is different)?

Answer 1.9

Thank you for raising this concern. We now address the effect of the location on the coefficient of performance in both method and result section.

We now write in the method section:

“On the consumption side, hourly demand for direct electricity of fans and compression unit are calculated based on the design capacity of the DAC plant and specific electricity requirements (see also Supplementary Note 1). Similarly, heat requirement of the DAC plant, for regeneration of saturated sorbent material, is transformed to electricity demand using a high-temperature heat pump. For heat pumps, the coefficient of performance (COP) indicates how much thermal energy is gained for each unit of electricity input ($kWh_{thermal} / kWh_{electricity}$). Regeneration of solid sorbent material requires temperatures of 100 °C. This level can be reached with state-of-the-art industrial, high temperature heat pumps⁴². The COP decreases for lower temperatures of the heat source. As the air temperature differs for each location, so does the COP. Using average air temperatures, we estimate COPs of 2.6 (Las Vegas, Nevada), 2.5 (Swakopmund, Namibia), and 2.3 (Munich, Germany) based on Arpagaus et al.⁴².”

And also address this point briefly in the result section:

“[...] Using solar irradiation data for Swakopmund (Namibia), which is similarly close to potential underground storage sites, leads to similar outcomes for the DAC utilization as Nevada. A slightly lower coefficient of performance (COP) for heat pumps used in Swakopmund, because of lower average temperatures than in Nevada, increases the energy consumption of the DAC system slightly.”

Comment 1.10

Page 22, Table 1. Could the authors elaborate on how this 100 km is decided? Does Arizona have saline aquifers (or equivalent) for CO₂ storage?

Answer 1.10

To provide an answer to your comment, we have reviewed relevant literature, and expanded on the requirements for underground CO₂ storage. Based on your comment, we find that locations in Nevada are more suitable than locations in Arizona. At the same time the solar irradiation profile of Arizona and the South of Nevada is quite similar. Thus we write in the method section that:

“For off-grid, PV powered systems, locations with high solar irradiation and substantial CO₂ underground storage potential are of interest. Wei et al.³⁹ and the U.S. Geological Survey⁴⁰ show the geographical distribution of suitable saline aquifer basins for underground carbon dioxide mineralization. To keep transportation distances of captured CO₂ to a minimum, the proposed DACS system is placed on top of a saline aquifer basin to reduce transport distances. With solar irradiation data from ref.⁴¹, we select Las Vegas (Nevada, USA), Swakopmund (Namibia), and Munich (Germany) as location for the analysis with suitable geological storage potential nearby³⁹. The underground storage of CO₂ must be durable, that is, without significant leakage, to effectively reduce atmospheric CO₂ levels and mitigate risks. Work on the security of underground CO₂ storage predicts that the cumulative CO₂ leakage will be 1.8 – 25% for a timespan of 10,000 years³⁰. This is equivalent to an annual leakage rate of 0.00018 – 0.0025%, which is very small compared to GHG emissions associated with the energy used for capture and storage of CO₂.”

And briefly in the result section:

“Las Vegas (Nevada) was selected due to the close location to suitable underground carbon dioxide storage, keeping transport distance to zero”.

To reflect our new approach, which places the DACS plant right on top of suitable saline storage formations, we have reduced the transport distance to 0 km in Table 1. The structure of our combined cost and environmental impact assessment model is still capable of including any other transport distance, if subsequent research is to take inspiration from our proposed method.

Table 1 Technological input data for off-grid DACS.

	Value	Unit	Sources
Design capacity	100,000	tCO ₂ / year	
Specific electricity demand	0.6	MWh _{el} / tCO ₂	10,24,25
Specific heat demand	3.33	MWh _{heat} / tCO ₂	20
Specific electricity demand for heat with heat pump (COP of 2.6 for Nevada)	1.28	MWh _{el} / tCO ₂	calculated
Specific electricity demand compression	0.1	MWh _{el} / tCO ₂	20,26,27
Overall specific electricity demand	1.98	MWh _{el} / tCO ₂	calculated
Distance to storage	0	km	
Adsorbent material consumption	7.5	kg / tCO ₂	20
Lifetime of DAC plant	12.5	years	
Lifetime of PV	25	years	
Lifetime of Battery storage	12.5	years	
Lifetime of Heat Pump	25	years	

Comment 1.11

Page 22, Table 2. Could the authors please add the dollar year to the table? Have these cost items been adjusted using CEPCI (for equipment) and CPI (for consumables). Doing the CEPCI and CPI corrections are rather important. Also, would there be labor cost?

Answer 1.11

Dear reviewer, there is no debate that accounting for inflation is always useful. We have, unfortunately, not thought about this ourselves initially. Thus, we have taken your advice and adjusted all cost data with CEPCI or CPI factors to reflect 2023 cost levels:

“Key data sources for technology, cost and environmental parameters are gathered through an extensive review of recent literature (see Supplementary Note 1). To reflect 2023 levels, cost data for consumables is adjusted with the consumer price index (CPI)⁵⁰ and for investments with the chemical engineering plant cost index (CEPCI)⁵¹. Thus, data in Table 1 and Table 2 represent a reference scenario for a state-of-the-art plant based on current LT solid sorbent technology.”

Reviewer #2

Comment 2.1

The manuscript presents an assessment of the cost and environmental impacts of an off-grid DACS (Direct Air Capture and Storage) system using solid sorbents. The work relies on literature data. It includes scenario analysis of future learning pathways and sensitivity analysis to pinpoint key parameters affecting the results. The conclusions are robust and supported by the analysis presented

Overall, I believe the manuscript is interesting, original (though primarily based on literature data) and thus worth publishing. I have also noticed that the Authors have addressed (to some extent) most of the comments of the previous round of review, which significantly improved the quality of the manuscript.

Answer 2.1

Thank you for this comment.

Comment 2.2

I am not clear whether the analysis considers leakages of CO₂ from transportation and permanent sequestration. As far as I am aware, literature estimates for both are low; but it is, in my opinion, worth at least discussing this, if not including both parameters in the sensitivity analysis.

Answer 2.2

We agree with your statement that CO₂ leakage was not addressed sufficiently in the previous version of the manuscript. Consequently, we looked into relevant literature and provided additional information about CO₂ leakage in the method section:

“Wei et al.³⁹ and the U.S. Geological Survey⁴⁰ show the geographical distribution of suitable saline aquifer basins for underground carbon dioxide mineralization. To keep transportation distances of captured CO₂ to a minimum, the proposed DACS system is placed on top of a saline aquifer basin to reduce transport distances. With solar irradiation data from ref.⁴¹, we select Las Vegas (Nevada, USA), Swakopmund (Namibia), and Munich (Germany) as location for the analysis with suitable geological storage potential nearby³⁹. The underground storage of CO₂ must be durable, that is, without significant leakage, to effectively reduce atmospheric CO₂ levels and mitigate risks. Work on the security of underground CO₂ storage predicts that the cumulative CO₂ leakage will be 1.8 – 25% for a timespan of 10,000 years³⁰. This is equivalent to an annual leakage rate of 0.00018 – 0.0025%, which is very small compared to GHG emissions associated with the energy used for capture and storage of CO₂.”

And restated a conservative leakage rate of 0.01% per year in the results sections to provide a perspective:

“For the net removal of 1 tCO₂ from the atmosphere, more than 1 tCO₂ must be captured and removed through the DACS to offset emissions associated with the process and leakage of CO₂ from storage. Note that annual leakage is less than 0.01% in well-managed geological CO₂ storage sites³⁰.”

Comment 2.3

The charts are nice, but not always easily understandable. For example, it is not clear to me what the grey dots (exponential curve) in Figure 3 represent, and how they differ from those vertical. I strongly suggest to include the full description of all components of the charts in the caption, and the symbol in the legend. Note that the same comment also applies to the other figures, e.g. Figure 3 and the black dots, which should all be checked and amended accordingly.

Answer 2.3

Thank you for this comment. Based on your recommendation, we have improved the visualization of all Figures and added a full description in the captions.

Fig. 1 Layout of solid-sorbent, off-grid DACS powered by PV-system, heat pump, and battery storage. Electricity generated by the PV-system is used for powering the air fans and compression unit of the direct air capture plant. Thermal energy for sorbent regeneration is provided by heat pumps. The battery storage allows for a smoother operation of the DAC plant as intermittency problems of the PV-system are (to some extent) addressed.

Fig. 2 Estimating utilization factors of off-grid DACS with various combinations of energy system and location. Electricity generated by the PV-system (yellow) depends on the nominal power of the PV-system (“P”) and the local solar irradiation profile. Excess electricity not required for powering the DACS system is used for charging the battery storage with nominal capacity (“C”) (red), which is discharged if solar irradiation decreases (green) to provide energy for continued DACS operation. Whenever PV-system and battery are not sufficient to meet the power requirement of the DAC, the capture rate (tCO₂ per hour) is reduced. **a** DACS located in Las Vegas (USA) with PV-power of 200 MW and battery storage of 300 MWh. **b** Same location but smaller PV-system (100 MW) and battery (100 MWh). **c** DACS system with 100 MW PV-system and 100 MWh battery in Munich (Germany), and **d** Swakopmund (Namibia).

Fig. 3 Annualized costs for off-grid DACs in Nevada (USA) with different combinations of PV-system and battery storage. Each dot represents the annualized costs (\$ per year) for a different combination of power of the PV-system (from 5 MW to 200 MW) and battery capacity (0 MWh to 300 MWh). Bars represent the lowest-cost option for utilization factors between 0% and 100%, corresponding to an annual gross removal capacity between 0 ktCO₂ and 100 ktCO₂ per year. Black dots represent combinations of the energy system with comparable utilization factors as the bars but higher annualized costs. Colors within bars reflect the contribution of depreciation of PV-system, battery storage, heat pump, and DAC plant, as well as variable costs of adsorbent replacement and permanent underground storage of captured CO₂.

Fig. 4 Levelized costs for gross and net removal of 1 tCO₂ with off-grid DACs in Nevada (USA). Bars represent the lowest (net) cost configurations of the PV-system and battery storage for different utilization factors. Colors of the bars reflect the contribution of each component (depreciation of battery, etc.) to total costs for the permanent removal of 1 tCO₂. Yellow parts of bar highlight the difference between costs for gross and net removal of 1 tCO₂. Costs for net removal of 1 tCO₂ include the carbon removal efficiency (which is <100%) for each configuration. Black dots represent the costs for net removal of 1 tCO₂ if the 45Q tax credit is subtracted from net costs.

Fig. 5 Global warming potential associated with net removal of 1 tCO₂ with off-grid DACS in Nevada (USA). Production and consumption of DACS components are associated with emission of greenhouse gases. Thus, for the net removal of 1 tCO₂ more than 1 tCO₂ must be captured and removed to offset emissions associated with the process. Bars represent the GWP for lowest net cost configurations for different utilization factors. Colors reflect the (positive) contribution of different components to GWP. Bars in light green represent the goal of net removing 1 tCO₂, with white bars accounting for the addition CO₂ capture and removal to offset (positive) GHG emissions of the capture and removal process.

Fig. 6 Environmental impacts associated with removal of 1 tCO₂ with off-grid DACs in Nevada (USA) equipped with a 100 MW PV-system and 300 MWh battery storage. Environmental impacts are shown for the 16 impact categories of the EF 3 LCA characterization model^{32,33}. Results are normalized against annual global totals in each impact category and weighed with factors from the European Commission³¹. Colors of the bars reflect the contribution of different components to impacts. Negative environmental impacts (meaning benefits) are achieved for the impact category of climate change, while positive environmental impacts (meaning burdens) result in all other impact categories.

Fig. 7 Normalized and weighed environmental impacts associated with removal of 1 tCO₂ with off-grid DACS in Nevada (USA). Sum of all positive environmental impacts after normalization and weighting³¹. Bars represent the lowest net costs combinations for different utilization factors, with colors indicating the contribution of each component to total environmental impact scores. Yellow parts of bar indicate the difference between net and gross removal of 1 tCO₂, accounting for CREs. Negative environmental impacts (benefits) for the impact category of climate change are -3.8×10^{-12} points per tCO₂ regardless of the utilization factor. Benefits of DACS in the climate change impact category are not included in this Figure.

Fig. 8 Economic-environmental dashboard for strategic learning scenarios of off-grid DACs. Results for the cost optimal DACS layout in Nevada (USA) with a 100 MW PV-system and 300 MWh battery represent the 100% benchmark for costs, GWP, and combined environmental impacts. Black dots show costs if the 45Q tax credit is included. Colors used in bars represent the contribution of components to total results. Yellow parts of bar indicate the difference between net and gross removal of 1 tCO₂, accounting for CREs.

Fig. 9 Sensitivity analysis on levelized costs for gross and net removal of 1 tCO₂ with off-grid DACS in Nevada (USA) equipped with a 100 MW PV-system and 300 MWh battery storage. Impact of 30% decrease (green) and 30% increase (red) of parameters on levelized costs for gross removal (left) and net removal (right) of 1 tCO₂ from the atmosphere.

Fig. 10 Structure of the utilization model. Design capacity, power of the PV-system, battery capacity, location, and power of the heat pump are key input parameters for the model, based upon which the utilization factor of the DACS system is estimated.

Fig. 11 Interaction between PV-electricity generation, battery storage and energy consumption by DACS system. For every hour in a representative year (2020) the model structure allows to calculate the hourly CO₂ capture rate of the DACS. Summing up results for one year gives estimates for the annual CO₂ removal and utilization factor.

Comment 2.4

The Authors should better describe the trade-offs associated with DACS, that is it has "negative" climate change impact but "positive" impacts in all other categories.

Answer 2.4

Your concern is of great importance. We have addressed it accordingly, which at the same time will help readers not already familiar with the DACS technology to get a better understanding of the trade-offs involved.

To improve clarity, we have redesigned Figure 5 to also include negative CO₂ emissions in the visualization and provide readers unfamiliar with DACS with a better understanding.

“Thus, for the net removal of 1 tCO₂ with a “100-300” system, a gross 1,115 tCO₂ must be captured and removed from the atmosphere of which the 0.115 tCO₂ offset the GWP of the process. In Fig. 5, GHG emissions are presented for lowest net-cost combinations.”

Fig. 12 Global warming potential associated with net removal of 1 tCO₂ with off-grid DACS in Nevada (USA). Production and consumption of DACS components are associated with emission of greenhouse gases. Thus, for the net removal of 1 tCO₂ more than 1 tCO₂ must be captured and removed to offset emissions associated with the process. Bars represent the GWP for lowest net cost configurations for different utilization factors. Colors reflect the (positive) contribution of different components to GWP. Bars in light green represent the goal of net removing 1 tCO₂, with white bars accounting for the addition CO₂ capture and removal to offset (positive) GHG emissions of the capture and removal process.

Further, based on your comment, we have included an additional Figure to address the situation that environmental benefits in one impact category (climate change) stand against environmental burdens in others, and adjusted the corresponding text:

“Apart from climate change, 15 other environmental impact categories are included in the life cycle assessment. To put the environmental impact categories into broader context, a normalization is done against the annual global impacts in each environmental impact category³¹. Further, relative weights are assigned to each impact category so that one final score is obtained for comparison of different options. Note, however, that the weights provided by the European Commission³¹ are subjective and do not yet represent an agreed upon standard. Fig. 6 shows normalized and weighed environmental impacts for the net removal of 1 tCO₂ with a 100 MW PV and 300 MWh battery storage in Nevada. Environmental benefits of -3.8×10^{-12} points are achieved in the impact category of climate change, which aligns with the general idea of DACS as a negative emissions technology. On

the other side, each ton CO₂ removed from atmosphere brings environmental burdens in other impact categories. Mineral resource scarcity, mostly associated with mining of critical raw materials used for the battery storage and PV system, but also adsorbent material and DAC plant, is of concern. An environmental burden of 6.0×10^{-12} points result for the “100-300” DACS layout. Put into context, 6×10^{-12} points mean that net-removal of 1 tCO₂ with this layout brings combined environmental impacts equivalent to $6 \times 10^{-10}\%$ of the global annual total. To provide an environmental assessment of each energy system combination, we use the sum of all positive environmental impacts.

Results for different energy system configurations are shown in Fig. 7. Other than for costs and GWP, lowest combined environmental impacts are achieved for lower utilization rates of 33% (“35-10”). This combination runs with only a small battery storage. For higher utilizations, however, the added environmental burden of increased battery capacity presents an issue. While the battery storage is not highly problematic from a cost and GWP perspective, using alternative energy storage options might be most suitable to reduce combined environmental impacts.”

Fig. 13 Environmental impacts associated with removal of 1 tCO₂ with off-grid DACs in Nevada (USA) equipped with a 100 MW PV-system and 300 MWh battery storage. Environmental impacts are shown for the 16 impact categories of the EF 3 LCA characterization model^{32,33}. Results are normalized against annual global totals in each impact category and weighed with factors from the European Commission³¹. Colors of the bars reflect the contribution of different components to impacts. Negative environmental impacts (meaning benefits) are achieved for the impact category of climate change, while positive environmental impacts (meaning burdens) result in all other impact categories.

Comment 2.5

The weighting factors used in the LCA study to obtain a single score describing the overall environmental performance are inherently value-based. This must be highlighted, particularly that different weighting factors may yield different results.

Answer 2.5

We address this concern by stating that:

“[...] To put the environmental impact categories into broader context, a normalization is done against the annual global impacts in each environmental impact category³¹. Further, relative weights are assigned to each impact category so that one final score is obtained for

comparison of different options. Note, however, that the weights provided by the European Commission³¹ are subjective and do not yet represent an agreed upon standard.”

The uncertainty around weighing is also reiterated in the discussion of optimizing environmental impacts of DACSs through learning scenarios. Here, we write:

“Finally, if all improvements were to occur simultaneously, net costs of \$ 369.5 per tCO₂ could be reached with a CRE of 93.6%. Including tax credits would bring costs down to \$216.5. These costs are substantially lower than the \$ 1,200 per tCO₂ currently charged by commercial DACS operators¹⁷. With combined environmental impact reduced to 3.1×10^{-12} points, the benefit of DACS for the impact category of climate change (which is -3.8×10^{-12} points) is larger than the sum of the environmental burden of the remaining 15 environmental impact categories. (Beware, however, of the uncertainty associated with normalization and weighing.)”

Comment 2.6

The concept of the "utilization" of the DAC plant is particularly important, but it is in my opinion not introduced with sufficient clarity in the manuscript. A reader who is not familiar with it may struggle understanding the results of the manuscript.

Answer 2.6

Thank you for this comment. We have rearranged the method section to introduce the utilization model earlier so that the relationship between utilization and cost modelling or LCA are easier to understand. Further we have rewritten the description of the utilization to guide readers:

“An off-grid DACS, powered by intermitted, renewable energy sources, is subject to fluctuations in electricity generation. For example, the energy generation with PV-systems depends strongly on the solar irradiation. If the sun does not shine, the PV-system does not generate electricity. Thus, a DACS system directly linked to the PV-system would have to stop or reduce operation whenever solar irradiation is low. Energy storage mediums, such as battery storage systems, can store electric energy for later use, thereby increasing the availability of the DACS system. However, adding battery capacity involves costs and environmental impacts. We conceive a basic structure of an algorithm to estimate the annual operating hours and captured CO₂ for numerous PV-systems, battery storage capacities, and locations. Interdependencies between technical parameters are shown in Fig. 10.”

We also include brief introduction of the utilization concept at the start of the result section to improve understanding of Figure 2. Further the caption of Figure 2 includes more basic information about the utilization:

“The DAC utilization model allows to estimate the hourly output of the DACS. With intermitted, renewable energy input, the DACS will not run at full capacity all the time. If the DAC runs at full capacity, it requires 22.6 MW electricity, of which 15.3 MW are used by heat

pumps generating $39.9 \text{ MW}_{\text{heat}}$ of thermal power. At full capacity the gross capture rate is 11.4 tCO_2 per hour, which is equivalent to $100,000 \text{ tCO}_2$ per year at 8760 operating hours. In Fig. 2, combinations of the energy system are denoted as “P-C” (“Power-Capacity”) for power of the PV system in MW and capacity of the battery in MWh.”

Fig. 14 Estimating utilization factors of off-grid DACS with various combinations of energy system and location. Electricity generated by the PV-system (yellow) depends on the nominal power of the PV-system (“P”) and the local solar irradiation profile. Excess electricity not required for powering the DACS system is used for charging the battery storage with nominal capacity (“C”) (red), which is discharged if solar irradiation decreases (green) to provide energy for continued DACS operation. Whenever PV-system and battery are not sufficient to meet the power requirement of the DAC, the capture rate (tCO_2 per hour) is reduced. **a** DACS located in Las Vegas (USA) with PV-power of 200 MW and battery storage of 300 MWh. **b** Same location but smaller PV-system (100 MW) and battery (100 MWh). **c** DACS system with 100 MW PV-system and 100 MWh battery in Munich (Germany), and **d** Swakopmund (Namibia).

And further:

“Based on an iterative approach, numerous combinations of photovoltaic system and battery storage are evaluated. Power of the PV system ranges from 5 MW to 200 MW, capacity of the battery storage from 0 MWh to 300 MWh. Similar to Fig. 2, utilization of the DAC system is calculated for each combination (in Fig. 3 on the x-axis). In addition, we calculate the annualized costs of DACS (ACOD) of each system configuration, shown in Fig. 3 as grey dots. A given utilization is reached with several different energy system combinations. For example, both “55-80” and “150-50” systems lead to ca. 50% utilization or (50,000 tCO₂ / year gross removed). But, with annualized costs of \$ 45.8 million, the “55-80” configuration is cheaper than the “150-50” combination with \$ 54.4 million. Consequently, bars in Fig. 4 represent the lowest cost layout for utilizations between 5% and 96% (5,000 to 96,000 tCO₂ per year removed).”

Comment 2.7

The sensitivity analysis is interesting, but I am not clear how this has been conducted. Have the Authors used plausible ranges for the parameters (which may differ for different parameters)? Or have they assumed the same range of variation? The interpretation of the results of the sensitivity analysis are heavily dependent on this.

Answer 2.7

We have redesigned the Figure showing results for the sensitivity analysis and also, based on your previous comment, improved the caption of this Figure:

Fig. 15 Sensitivity analysis on leveled costs for gross and net removal of 1 tCO₂ with off-grid DACS in Nevada (USA) equipped with a 100 MW PV-system and 300 MWh battery storage. Impact of 30% decrease (green) and 30% increase (red) of parameters on leveled costs for gross removal (left) and net removal (right) of 1 tCO₂ from the atmosphere.

In addition, we rewrote the text corresponding to the sensitivity analysis to address your comment:

“A brief sensitivity analysis, conducted for a flat 30% increase or decrease in key technical parameters, shows interest rate, capex of DAC, and adsorbent costs as most

relevant factors, see Fig. 9. Therefore, providing DACS companies with access to funding with 7% interest rate rather than 10%, reduces costs by \$ >90 per tCO₂, pointing to possible strategies for policy subsidies. Reducing the GWP of the adsorbent material or the battery storage has no effect on the gross costs of DACS. However, for net costs, reduced GWP associated with the production of the PV system or battery storage, has the potential to reduce net costs by \$ 15. This might provide a basis for negotiation between manufacturers of adsorbent materials and DACS operators.”

REVIEWERS' COMMENTS:

Reviewer #1 (Remarks to the Author):

After carefully examining the revised manuscript, I think the authors have addressed my comments appropriately. Therefore, I recommend that this manuscript be published in its current state.

Reviewer #2 (Remarks to the Author):

I am happy with the changes made, and I have no more comments.